# ON THE LIMITATIONS OF FIRST-ORDER APPROXIMATION IN GAN DYNAMICS

## ABSTRACT

Generative Adversarial Networks (GANs) have been proposed as an approach to learning generative models. While GANs have demonstrated promising performance on multiple vision tasks, their learning dynamics are not yet well understood, neither in theory nor in practice. In particular, the work in this domain has been focused so far only on understanding the properties of the *stationary* solutions that this dynamics might converge to, and of the behavior of that dynamics in this solutions' immediate neighborhood.

To address this issue, in this work we take a first step towards a principled study of the GAN dynamics itself. To this end, we propose a model that, on one hand, exhibits several of the common problematic convergence behaviors (e.g., vanishing gradient, mode collapse, diverging or oscillatory behavior), but on the other hand, is sufficiently simple to enable *rigorous* convergence analysis.

This methodology enables us to exhibit an interesting phenomena: a GAN with an optimal discriminator provably converges, while guiding the GAN training using only a first order approximation of the discriminator leads to unstable GAN dynamics and mode collapse. This suggests that such usage of the first order approximation of the discriminator, which is a de-facto standard in all the existing GAN dynamics, might be one of the factors that makes GAN training so challenging in practice. Additionally, our convergence result constitutes the first rigorous analysis of a dynamics of a concrete parametric GAN.

## 1 INTRODUCTION

Generative modeling is a fundamental learning task of growing importance. As we apply machine learning to increasingly sophisticated problems, we often aim to learn functions with an output domain that is significantly more complex than simple class labels. Common examples include image "translation" (Isola et al., 2017), speech synthesis (van den Oord et al., 2016), and robot trajectory prediction (Finn et al., 2016). Due to progress in deep learning, we now have access to powerful architectures that can represent generative models over such complex domains. However, training these generative models is a key challenge. Simpler learning problems such as classification have a clear notion of "right" and "wrong," and the approaches based on minimizing the corresponding loss functions have been tremendously successful. In contrast, training a generative model is far more nuanced because it is often unclear how "good" a sample from the model is.

Generative Adversarial Networks (GANs) have recently been proposed to address this issue (Goodfellow et al., 2014). In a nutshell, the key idea of GANs is to learn *both* the generative model and the loss function at the same time. The resulting training dynamics are usually described as a game between a generator (the generative model) and a discriminator (the loss function). The goal of the generator is to produce realistic samples that fool the discriminator, while the discriminator is trained to distinguish between the true training data and samples from the generator. GANs have shown promising results on a variety of tasks, and there is now a large body of work that explores the power of this framework (Goodfellow, 2017).

Unfortunately, reliably training GANs is a challenging problem that often hinders further research in this area. Practitioners have encountered a variety of obstacles such as vanishing gradients, mode collapse, and diverging or oscillatory behavior (Goodfellow, 2017). At the same time, the theoretical underpinnings of GAN dynamics are not yet well understood. To date, there were no convergence

proofs for GAN models, even in very simple settings. As a result, the root cause of frequent failures of GAN dynamics in practice remains unclear.

In this paper, we take a first step towards a principled understanding of GAN *dynamics*. Our general methodology is to propose and examine a problem setup that exhibits all common failure cases of GAN dynamics while remaining sufficiently simple to allow for a rigorous analysis. Concretely, we introduce and study the GMM-GAN: a variant of GAN dynamics that captures learning a mixture of two univariate Gaussians. We first show experimentally that standard gradient dynamics of the GMM-GAN often fail to converge due to mode collapse or oscillatory behavior. Interestingly, this also holds for techniques that were recently proposed to improve GAN training such as unrolled GANs (Metz et al., 2017). In contrast, we then show that GAN dynamics with an *optimal* discriminator *do* converge, both experimentally and *provably*. To the best of our knowledge, our theoretical analysis of the GMM-GAN is the first global convergence proof for parametric and non-trivial GAN dynamics.

Our results show a clear dichotomy between the dynamics arising from applying simultaneous gradient descent and the one that is able to use an optimal discriminator. The GAN with optimal discriminator provably converges from (essentially) *any* starting point. On the other hand, the simultaneous gradient GAN empirically often fails to converge, even when the discriminator is allowed many more gradient steps than the generator. These findings go against the common wisdom that first order methods are sufficiently strong for all deep learning applications. By carefully inspecting our models, we are able to pinpoint some of the causes of this, and we highlight a phenomena we call *discriminator collapse* which often causes first order methods to fail in our setting.

## 2 GENERATIVE ADVERSARIAL DYNAMICS

Generative adversarial networks are commonly described as a two player game (Goodfellow et al., 2014). Given a true distribution $P$, a set of generators $\mathcal{G} = \{G_u, u \in \mathcal{U}\}$, a set of discriminators $\mathcal{D} = \{D_v, v \in \mathcal{V}\}$, and a monotone measuring function $m : \mathbb{R} \to \mathbb{R}$, the objective of GAN training is to find a generator $u$ in

$$\arg\min_{u \in \mathcal{U}} \max_{v \in \mathcal{V}} \mathbb{E}_{x \sim P}[m(D_v(x))] + \mathbb{E}_{x \sim G_u}[m(1 - D_v(x))] . \tag{1}$$

In other words, the game is between two players called the generator and discriminator, respectively. The goal of the discriminator is to distinguish between samples from the generator and the true distribution. The goal of the generator is to fool the discriminator by generating samples that are similar to the data distribution.

By varying the choice of the measuring function and the set of discriminators, one can capture a wide variety of loss functions. Typical choices that have been previously studied include the KL divergence and the Wasserstein distance (Goodfellow et al., 2014; Arjovsky et al., 2017). This formulation can also encode other common objectives: most notably, as we will show, the total variation distance.

To optimize the objective (1), the most common approaches are variants of simultaneous gradient descent on the generator $u$ and the discriminator $v$. But despite its attractive theoretical grounding, GAN training is plagued by a variety of issues in practice. Two major problems are *mode collapse* and *vanishing gradients*. Mode collapse corresponds to situations in which the generator only learns a subset (a few modes) of the true distribution $P$ (Goodfellow, 2017; Arora & Zhang, 2017). For instance, a GAN trained on an image modeling task would only produce variations of a small number of images. Vanishing gradients (Arjovsky et al., 2017; Arjovsky & Bottou, 2017; Arora et al., 2017) are, on the other hand, a failure case where the generator updates become vanishingly small, thus making the GAN dynamics not converge to a satisfying solution. Despite many proposed explanations and approaches to solve the vanishing gradient problem, it is still often observed in practice (Goodfellow, 2017).

### 2.1 TOWARDS A PRINCIPLED UNDERSTANDING OF GAN DYNAMICS

GANs provide a powerful framework for generative modeling. However, there is a large gap between the theory and practice of GANs. Specifically, to the best of the authors' knowledge, all theoretical studies of GAN dynamics for parametric models simply consider global optima and stationary points of the dynamics, and there has been no rigorous study of the actual GAN dynamics. In practice,

GANs are always optimized using first order methods, and the current theory of GANs cannot tell us whether or not these methods converge to a meaningful solution. This raises a natural question, also posed as an open problem in (Goodfellow, 2017):

Our theoretical understanding of GANs is still fairly poor. In particular, to the best of the authors' knowledge, all existing analyzes of GAN dynamics for parametric models simply consider global optima and stationary points of the dynamics. There has been no rigorous study of the actual GAN dynamics, except studying it in the immediate neighborhood of such stationary points (Nagarajan & Kolter, 2017). This raises a natural question:

*Can we understand the convergence behavior of GANs?*

This question is difficult to tackle for many reasons. One of them is the non-convexity of the GAN objective/loss function, and of the generator and discriminator sets. Another one is that, in practice, GANs are always optimized using first order methods. That is, instead of following the "ideal" dynamics that has both the generator and discriminator always perform the optimal update, we just approximate such updates by a sequence of gradient steps. This is motivated by the fact that computing such optimal updates is, in general, algorithmically intractable, and adds an additional layer of complexity to the problem.

In this paper, we want to change this state of affairs and initiate the study of GAN dynamics from an algorithmic perspective. Specifically, we pursue the following question:

*What is the impact of using first order approximation on the convergence of GAN dynamics?*

Concretely, we focus on analyzing the difference between two GAN dynamics: a "first order" dynamics, in which both the generator and discriminator use first order updates; and an "optimal discriminator" dynamics, in which only the generator uses first order updates but the discriminator always makes an optimal update. Even the latter, simpler dynamics has proven to be challenging to understand. Even the question of whether using the optimal discriminator updates is the right approach has already received considerable attention. In particular, (Arjovsky & Bottou, 2017) present theoretical evidence that using the optimal discriminator at each step may not be desirable in certain settings (although these settings are very different to the one we consider in this paper).

We approach our goal by defining a simple GAN model whose dynamics, on one hand, captures many of the difficulties of real-world GANs but, on the other hand, is still simple enough to make analysis possible. We then rigorously study our questions in the context of this model. Our intention is to make the resulting understanding be the first step towards crystallizing a more general picture.

## 3  A Simple Model for Studying GAN Dynamics

Perhaps a tempting starting place for coming up with a simple but meaningful set of GAN dynamics is to consider the generators being univariate Gaussians with fixed variance. Indeed, in the supplementary material we give a short proof that simple GAN dynamics always converge for this class of generators. However, it seems that this class of distributions is insufficiently expressive to exhibit many of the phenomena such as mode collapse mentioned above. In particular, the distributions in this class are all unimodal, and it is unclear what mode collapse would even mean in this context.

**Generators.**  The above considerations motivate us to make our model slightly more complicated. We assume that the true distribution and the generator distributions are all mixtures of two univariate Gaussians with unit variance, and uniform mixing weights. Formally, our generator set is $\mathcal{G}$, where

$$\mathcal{G} = \left\{ \frac{1}{2}\mathcal{N}(\mu_1, 1) + \frac{1}{2}\mathcal{N}(\mu_2, 1) \mid \mu_1, \mu_2 \in \mathbb{R} \right\} . \tag{2}$$

For any $\mu \in \mathbb{R}^2$, we let $G_\mu(x)$ denote the distribution in $\mathcal{G}$ with means at $\mu_1$ and $\mu_2$. While this is a simple change compared to a single Gaussian case, it makes a large difference in the behavior of the dynamics. In particular, many of the pathologies present in real-world GAN training begin to appear.

**Loss function.**  While GANs are usually viewed as a generative framework, they can also be viewed as a general method for density estimation. We want to set up learning an unknown generator $G_{\mu^*} \in \mathcal{G}$ as a generative adversarial dynamics. To this end, we must first define the loss function

for the density estimation problem. A well-studied goal in this setting is to recover $G_{\mu^*}(x)$ in total variation (also known as $L^1$ or statistical) distance, where the total variation distance between two distributions $P, Q$ is defined as

$$d_{\text{TV}}(P, Q) = \frac{1}{2} \int_\Omega |P(x) - Q(x)| dx = \max_A P(A) - Q(A) \;, \tag{3}$$

where the maximum is taken over all measurable events $A$.

Such finding the best-fit distribution in total variation distance can indeed be naturally phrased as generative adversarial dynamics. Unfortunately, for arbitrary distributions, this is algorithmically problematic, simply because the set of discriminators one would need is intractable to optimize over.

However, for distributions that are structurally simple, like mixtures of Gaussians, it turns out we can consider a much simpler set of discriminators. In Appendix B.1 in the supplementary material, motivated by connections to VC theory, we show that for two generators $G_{\mu_1}, G_{\mu_2} \in \mathcal{G}$, we have

$$d_{\text{TV}}(G_{\mu_1}, G_{\mu_2}) = \max_{E = I_1 \cup I_2} G_{\mu_1}(E) - G_{\mu_2}(E) \;, \tag{4}$$

where the maxima is taken over two disjoint intervals $I_1, I_2 \subseteq \mathbb{R}$. In other words, instead of considering the difference of measure between the two generators $G_{\mu_1}, G_{\mu_2}$ on arbitrary events, we may restrict our attention to unions of two disjoint intervals in $\mathbb{R}$. This is a special case of a well-studied distance measure known as the $\mathcal{A}_k$-distance, for $k = 2$ (Devroye & Lugosi, 2012; Chan et al., 2014). Moreover, this class of subsets has a simple parametric description.

**Discriminators.** Now, the above discussion motivates our definition of discriminators to be

$$\mathcal{D} = \{\mathbb{I}_{[\ell_1, r_1]} + \mathbb{I}_{[\ell_2, r_2]} \mid \ell, r \in \mathbb{R}^2 \text{ s.t. } \ell_1 \le r_1 \le \ell_2 \le r_2\} \;. \tag{5}$$

In other words, the set of discriminators is taken to be the set of indicator functions of sets which can be expressed as a union of at most two disjoint intervals. With this definition, finding the best fit in total variation distance to some unknown $G_{\mu^*} \in \mathcal{G}$ is equivalent to finding $\widehat{\mu}$ minimizing

$$\widehat{\mu} = \arg\min_\mu \max_{\ell, r} L(\mu, \ell, r) \;, \text{ where } L(\mu, \ell, r) = \mathbb{E}_{x \sim G_{\mu^*}}[D(x)] + \mathbb{E}_{x \sim G_\mu}[1 - D(x)] \tag{6}$$

is a smooth function of all three parameters (see the supplementary material for details).

**Dynamics.** The objective in (6) is easily amenable to optimization at parameter level. A natural approach for optimizing this function would be to define $G(\widehat{\mu}) = \max_{\ell, r} L(\widehat{\mu}, \ell, r)$, and to perform (stochastic) gradient descent on this function. This corresponds to, at each step, finding the the optimal discriminator, and updating the current $\widehat{\mu}$ in that direction. We call these dynamics the *optimal discriminator dynamics*. Formally, given $\widehat{\mu}^{(0)}$ and a stepsize $\eta_g$, and a true distribution $G_{\mu^*} \in \mathcal{G}$, the optimal discriminator dynamics for $G_{\mu^*}, \mathcal{G}, \mathcal{D}$ starting at $\widehat{\mu}^{(0)}$ are given iteratively as

$$\ell^{(t)}, r^{(t)} = \arg\max_{\ell, r} L(\widehat{\mu}^{(t)}, \ell, r) \;, \quad \widehat{\mu}^{(t+1)} = \widehat{\mu}^{(t)} - \eta_g \nabla_\mu L(\widehat{\mu}^{(t)}, \ell^{(t)}, r^{(t)}) \;, \tag{7}$$

where the maximum is taken over $\ell, r$ which induce two disjoint intervals.

For more complicated generators and discriminators such as neural networks, these dynamics are computationally difficult to perform. Therefore, instead of the updates as in (7), one resorts to simultaneous gradient iterations on the generator and discriminator. These dynamics are called the *first order dynamics*. Formally, given $\widehat{\mu}^{(0)}, \ell^{(0)}, r^{(0)}$ and a stepsize $\eta_g, \eta_d$, and a true distribution $G_{\mu^*} \in \mathcal{G}$, the first order dynamics for $G_{\mu^*}, \mathcal{G}, \mathcal{D}$ starting at $\widehat{\mu}^{(0)}$ are specified as

$$\widehat{\mu}^{(t+1)} = \widehat{\mu}^{(t)} - \eta_g \nabla_\mu L(\widehat{\mu}^{(t)}, \ell^{(t)}, r^{(t)}) \tag{8}$$

$$r^{(t+1)} = r^{(t)} + \eta_d \nabla_r L(\widehat{\mu}^{(t)}, \ell^{(t)}, r^{(t)}) \;, \quad \ell^{(t+1)} = \ell^{(t)} + \eta_d \nabla_\ell L(\widehat{\mu}^{(t)}, \ell^{(t)}, r^{(t)}) \;. \tag{9}$$

Even for our relatively simple setting, the first order dynamics can exhibit a variety of behaviors, depending on the starting conditions of the generators and discriminators. In particular, in Figure 1, we see that depending on the initialization, the dynamics can either converge to optimality, exhibit a primitive form of mode collapse, where the two generators collapse into a single generator, or converge to the wrong value, because the gradients vanish. This provides empirical justification for our model, and shows that these dynamics are complicated enough to model the complex behaviors which real-world GANs exhibit. Moreover, as we show in Section 5 below, these behaviors are not just due to very specific pathological initial conditions: indeed, when given random initial conditions, the first order dynamics still more often than not fail to converge.

**Parametrization**   We note here that there could be several potential GAN dynamics to consider here. Each one resulting from slightly different parametrization of the total variation distance. For instance, a completely equivalent way to define the total variation distance is

$$d_{\text{TV}}(P, Q) = \max_A |P(A) - Q(A)| , \tag{10}$$

which does not change the value of the variational distance, but does change the induced dynamics. We do not focus on these induced dynamics in this paper since they do not exactly fit within the traditional GAN framework, i.e. it is not of the form (1) (see Appendix C). Nevertheless, it is an interesting set of dynamics and it is a natural question whether similar phenomena occur in these dynamics. In Appendix C, we show the the optimal discriminator dynamics are unchanged, and the induced first order dynamics have qualitatively similar behavior to the ones we consider in this paper. This also suggests that the phenomena we exhibit might be more fundamental.

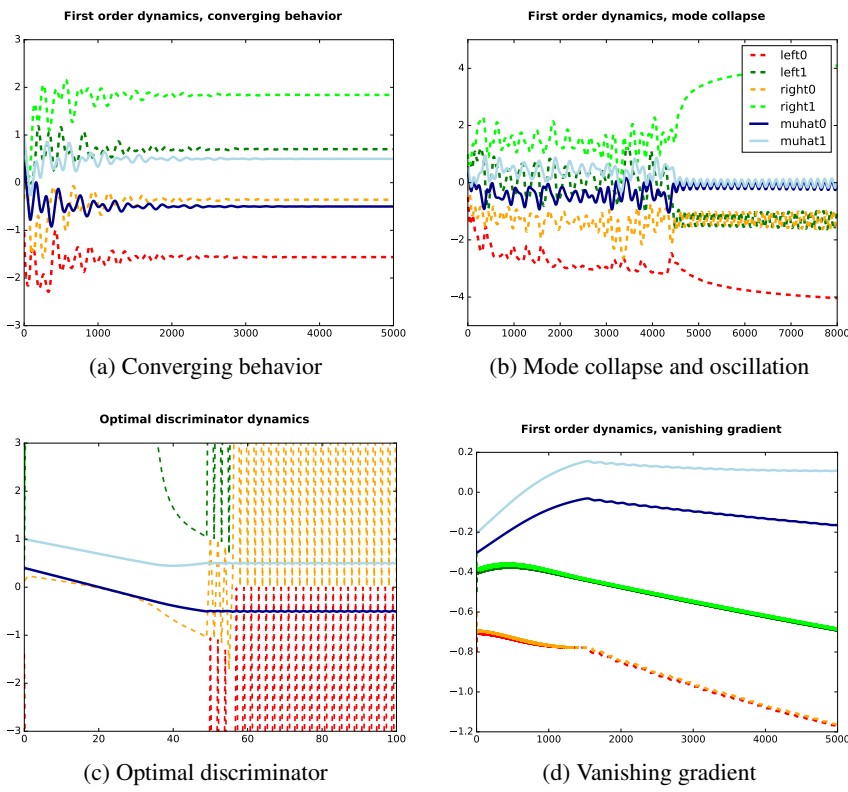

Figure 1: A selection of different GAN behaviors. In all plots the true distribution was $G_{\mu^*}$ with $\mu^* = (-0.5, 0.5)$, and step size was taken to be $0.1$. The solid lines represent the two coordinates of $\widehat{\mu}$, and the dotted lines represent the discriminator intervals. In order: (a) first order dynamics with initial conditions that converge to the true distribution. (b) First order dynamics with initial conditions that exhibit wild oscillation before mode collapse. (c) Optimal discriminator dynamics. (d) First order dynamics that exhibit vanishing gradients and converge to the wrong distribution. Observe that the optimal discriminator dynamics converge, and then the discriminator varies wildly, because the objective function is not differentiable at optimality. Despite this it remains roughly at optimality from step to step.

## 4   OPTIMAL DISCRIMINATOR VS. FIRST ORDER DYNAMICS

We now describe our results in more detail. We first consider the dynamics induced by the optimal discriminator. Our main theoretical result is[1]:

---

[1]We actually analyze a minor variation on the optimal discriminator dynamics. In particular, we do not rule out the existence of a measure zero set on which the dynamics are ill-behaved. Thus, we will analyze the optimal

**Theorem 4.1.** *Fix $\delta > 0$ sufficiently small and $C > 0$. Let $\mu^* \in \mathbb{R}^2$ so that $|\mu_i^*| \leq C$, and $|\mu_1^* - \mu_2^*| \geq \delta$. Then, for all initial points $\widehat{\mu}^{(0)}$ so that $|\widehat{\mu}_i^{(0)}| \leq C$ for all $i$ and so that $|\widehat{\mu}_1^{(0)} - \widehat{\mu}_2^{(0)}| \geq \delta$, if we let $\eta = \mathrm{poly}(1/\delta, e^{-C^2})$ and $T = \mathrm{poly}(1/\delta, e^{-C^2})$, then if $\widehat{\mu}^{(T)}$ is specified by the optimal discriminator dynamics, we have $d_{\mathrm{TV}}(G_{\mu^*}, G_{\widehat{\mu}^{(T)}}) \leq \delta$.*

In other words, if the $\mu^*$ are bounded by a constant, and not too close together, then in time which is polynomial in the inverse of the desired accuracy $\delta$ and $e^{-C^2}$, where $C$ is a bound on how far apart the $\mu^*$ and $\widehat{\mu}$ are, the optimal discriminator dynamics converge to the ground truth in total variation distance. Note that the dependence on $e^{-C^2}$ is necessary, as if the $\widehat{\mu}$ and $\mu^*$ are initially very far apart, then the initial gradients for the $\widehat{\mu}$ will necessarily be of this scale as well.

On the other hand, we provide simulation results that demonstrate that first order updates, or more complicated heuristics such as unrolling, all fail to consistently converge to the true distribution, even under the same sorts of conditions as in Theorem 4.1. In Figure 1, we gave some specific examples where the first order dynamics fail to converge. In Section 5 we show that this sort of divergence is common, even with random initializations for the discriminators. In particular, the probability of convergence is generally much lower than 1, for both the regular GAN dynamics, and unrolling. In general, we believe that this phenomena should occur for *any* natural first order dynamics for the generator. In particular, one barrier we observed for any such dynamics is something we call *discriminator collapse*, that we describe in Appendix A.

### 4.1 Analyzing the Optimal Discriminator Dynamics

We provide now a high level overview of the proof of Theorem 4.1. The key element we will need in our proof is the ability to quantify the progress our updates make on converging towards the optimal solution. This is particularly challenging as our objective function is neither convex nor smooth. The following lemma is our main tool for achieving that. Roughly stated, it says that for any Lipschitz function, even if it is non-convex and *non-smooth*, as long as the change in its derivative is smaller in magnitude than the value of the derivative, gradient descent makes progress on the function value. Note that this condition is much weaker than typical assumptions used to analyze gradient descent.

**Lemma 4.2.** *Let $g : \mathbb{R}^k \to \mathbb{R}$ be a Lipschitz function that is differentiable at some fixed $x \in \mathbb{R}^k$. For some $\eta > 0$, let $x' = x - \eta\nabla f(x)$. Suppose there exists $c < 1$ so that almost all $v \in L(x, x')$, where $L(x, y)$ denotes the line between $x$ and $y$, $g$ is differentiable, and moreover, we have $\|\nabla g(x) - \nabla g(v)\|_2 \leq c\|\nabla g(x)\|_2$. Then $g(x') - g(x) \leq -\eta(1 - c)\|\nabla g(x)\|_2^2$.*

Here, we will use the convention that $\mu_1^* \leq \mu_2^*$, and during the analysis, we will always assume for simplicity of notation that $\widehat{\mu}_1 \leq \widehat{\mu}_2$. Also, in what follows, let $f(\widehat{\mu}) = f_{\mu^*}(\widehat{\mu}) = d_{\mathrm{TV}}(G_{\widehat{\mu}}, G_{\mu^*})$ and $F(\widehat{\mu}, x) = G_{\mu^*}(x) - G_{\widehat{\mu}}(x)$ be the objective function and the difference of the PDFs between the true distribution and the generator, respectively.

For any $\delta > 0$, define the sets

$$\mathrm{Rect}(\delta) = \{\widehat{\mu} : |\widehat{\mu}_i - \mu_j^*| < \delta \text{ for some } i, j\} \quad , \quad \mathrm{Opt}(\delta) = \{\widehat{\mu} : |\widehat{\mu}_i - \mu_i^*| < \delta \text{ for all } i\} .$$

to be the set of parameter values which have at least one parameter which is not too far from optimality, and the set of parameter values so that all parameter values are close. We also let $B(C)$ denote the box of sidelength $C$ around the origin, and we let $\mathrm{Sep}(\gamma) = \{v \in \mathbb{R}^2 : |v_1 - v_2| > \gamma\}$ be the set of parameter vectors which are not too close together.

Our main work lies within a set of lemmas which allow us to instantiate the bounds in Lemma 4.2. We first show a pair of lemmas which show that, explicitly excluding bad cases such as mode collapse, our dynamics satisfy the conditions of Lemma 4.2. We do so by establishing a strong (in fact, nearly constant) lower bound on the gradient when we are fairly away from optimality (Lemma 4.3). Then, we show a relatively weak bound on the smoothness of the function (Lemma 4.4), but which is sufficiently strong in combination with Lemma 4.3 to satisfy Lemma 4.2. Finally, we rule

---

discriminator dynamics after adding an arbitrarily small amount of Gaussian noise. It is clear that by taking this noise to be sufficiently small (say exponentially small) then we avoid this pathological set with probability 1, and moreover the noise does not otherwise affect the convergence analysis at all. For simplicity, we will ignore this issue for the rest of the paper.

out the pathological cases we explicitly excluded earlier, such as mode collapse or divergent behavior (Lemmas 4.5 and 4.6). Putting all these together appropriately yields the desired statement. Our first lemma is a lower bound on the gradient value:

**Lemma 4.3.** *Fix $C \geq 1 \geq \gamma \geq \delta > 0$. Suppose $\widehat{\mu} \notin \mathrm{Rect}(0)$, and suppose $\mu^*, \widehat{\mu} \in B(C)$ and $\mu^* \in \mathrm{Sep}(\gamma), \widehat{\mu} \in \mathrm{Sep}(\delta)$. There is some $K = \Omega(1) \cdot (\delta e^{-C^2}/C)^{O(1)}$ so that $\|\nabla f_{\mu^*}(\widehat{\mu})\|_2 \geq K$.*

The above lemma statement is slightly surprising at first glance. It says that the gradient is never 0, which would suggest there are no local optima at all. To reconcile this, one should note that the gradient is not continuous (defined) everywhere.

The second lemma states a bound on the smoothness of the function:

**Lemma 4.4.** *Fix $C \geq 1$ and $\gamma \geq \delta > 0$ so that $\delta$ is sufficiently small. Let $\mu^*, \widehat{\mu}, \widehat{\mu}'$ be such that $L(\widehat{\mu}, \widehat{\mu}') \cap \mathrm{Opt}(\delta) = \varnothing$, $\mu^* \in \mathrm{Sep}(\gamma)$, $\widehat{\mu}', \widehat{\mu} \in \mathrm{Sep}(\delta)$, and $\mu^*, \widehat{\mu}, \widehat{\mu}' \in B(C)$. Let $K = \Omega(1) \cdot (\delta e^{-C^2}/C)^{O(1)}$ be the $K$ for which Lemma 4.3 holds with those parameters. If we have $\|\widehat{\mu}' - \widehat{\mu}\|_2 \leq \Omega(1) \cdot (\delta e^{-C^2}/C)^{O(1)}$ for appropriate choices of constants on the RHS, we get*

$$\|\nabla f_{\mu^*}(\widehat{\mu}') - \nabla f_{\mu^*}(\widehat{\mu})\|_2 \leq K/2 \leq \|\nabla f_{\mu^*}(\widehat{\mu})\|_2/2.$$

These two lemmas almost suffice to prove progress as in Lemma 4.2, however, there is a major caveat. Specifically, Lemma 4.4 needs to assume that $\widehat{\mu}$ and $\widehat{\mu}'$ are sufficiently well-separated, and that they are bounded. While the $\widehat{\mu}_i$ start out separated and bounded, it is not clear that it does not mode collapse or diverge off to infinity. However, we are able to rule these sorts of behaviors out. Formally:

**Lemma 4.5** (No mode collapse). *Fix $\gamma > 0$, and let $\delta$ be sufficiently small. Let $\eta \leq \delta/C$ for some $C$ large. Suppose $\mu^* \in \mathrm{Sep}(\gamma)$. Then, if $\widehat{\mu} \in \mathrm{Sep}(\delta)$, and $\widehat{\mu}' = \widehat{\mu} - \eta \nabla f_{\mu^*}(\widehat{\mu})$, we have $\widehat{\mu}' \in \mathrm{Sep}(\delta)$.*

**Lemma 4.6** (No diverging to infinity). *Let $C > 0$ be sufficiently large, and let $\eta > 0$ be sufficiently small. Suppose $\mu^* \in B(C)$, and $\widehat{\mu} \in B(2C)$. Then, if we let $\widehat{\mu}' = \widehat{\mu} - \eta \nabla f_{\mu^*}(\widehat{\mu})$, then $\widehat{\mu}' \in B(2C)$.*

Together, these four lemmas together suffice to prove Theorem 4.1 by setting parameters appropriately. We refer the reader to Appendix D for more details including the proofs.

## 5 EXPERIMENTS

To illustrate more conclusively that the phenomena demonstrated in Figure 1 are not particularly rare, and that first order dynamics do often fail to converge, we also conducted the following heatmap experiments. We set $\mu^* = (-0.5, 0.5)$ as in Figure 1. We then set a grid for the $\widehat{\mu}$, so that each coordinate is allowed to vary from $-1$ to $1$. For each of these grid points, we randomly chose a set of initial discriminator intervals, and ran the first order dynamics for 3000 iterations, with constant stepsize 0.3. We then repeated this 120 times for each grid point, and plotted the probability that the generator converged to the truth, where we say the generator converged to the truth if the TV distance between the generator and optimality is $< 0.1$. The choice of these parameters was somewhat arbitrary, however, we did not observe any qualitative difference in the results by varying these numbers, and so we only report results for these parameters. We also did the same thing for the optimal discriminator dynamics, and for unrolled discriminator dynamics with 5 unrolling steps, as described in (Metz et al., 2017), which attempt to match the optimal discriminator dynamics.

The results of the experiment are given in Figure 2. We see that all three methods fail when we initialize the two generator means to be the same. This makes sense, since in that regime, the generator starts out mode collapsed and it is impossible for it to un-"mode collapse", so it cannot fit the true distribution well. Ignoring this pathology, we see that the optimal discriminator otherwise always converges to the ground truth, as our theory predicts. On the other hand, both regular first order dynamics and unrolled dynamics often times fail, although unrolled dynamics do succeed more often than regular first order dynamics. This suggests that the pathologies in Figure 1 are not so rare, and that these first order methods are quite often unable to emulate optimal discriminator dynamics.

## 6 RELATED WORK

GANs have received a tremendous amount of attention over the past two years (Goodfellow, 2017). Hence we only compare our results to the most closely related papers here.

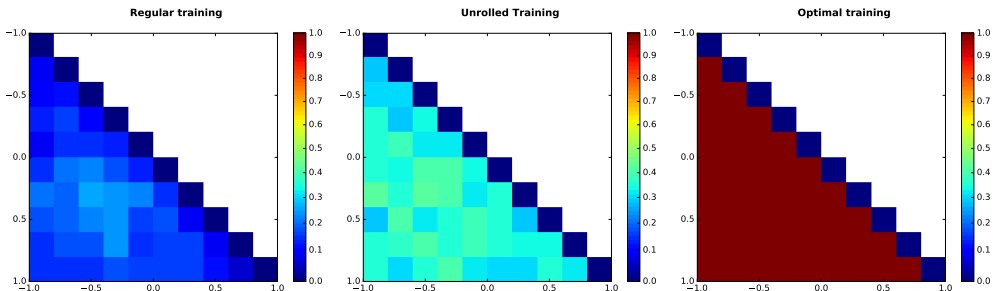

Figure 2: Heatmap of success probability for random discriminator initialization for regular GAN training, unrolled GAN training, and optimal discriminator dynamics.

The recent paper (Arora et al., 2017) studies generalization aspects of GANs and the existence of equilibria in the two-player game. In contrast, our paper focuses on the *dynamics* of GAN training. We provide the first rigorous proof of global convergence and show that a GAN with an optimal discriminator always converges to an approximate equilibrium.

One recently proposed method for improving the convergence of GAN dynamics is the unrolled GAN (Metz et al., 2017). The paper proposes to "unroll" multiple discriminator gradient steps in the generator loss function. The authors argue that this improves the GAN dynamics by bringing the discriminator closer to an optimal discriminator response. Our experiments show that this is not a perfect approximation: the unrolled GAN still fails to converge in multiple initial configurations (however, it does converge more often than a "vanilla" one-step discriminator).

The authors of (Arjovsky & Bottou, 2017) also take a theoretical view on GANs. They identify two important properties of GAN dynamics: (i) Absolute continuity of the population distribution, and (ii) overlapping support between the population and generator distribution. If these conditions do not hold, they show that the GAN dynamics fail to converge in some settings. However, they do not prove that the GAN dynamics *do* converge under such assumptions. We take a complementary view: we give a convergence proof for a concrete GAN dynamics. Moreover, our model shows that absolute continuity and support overlap are not the only important aspects in GAN dynamics: although our distributions clearly satisfy both of their conditions, the first-order dynamics still fail to converge.

The paper (Nagarajan & Kolter, 2017) studies the stability of equilibria in GAN training. In contrast to our work, the results focus on *local* stability while we establish *global* convergence results. Moreover, their theorems rely on fairly strong assumptions. While the authors give a concrete model for which these assumptions are satisfied (the linear quadratic Gaussian GAN), the corresponding target and generator distributions are *unimodal*. Hence this model cannot exhibit mode collapse. We propose the GMM-GAN specifically because it is rich enough to exhibit mode collapse.

The recent work (Grnarova et al., 2017) views GAN training through the lens of online learning. The paper gives results for the game-theoretic minimax formulation based on results from online learning. The authors give results that go beyond the convex-concave setting, but do not address generalization questions. Moreover, their algorithm is not based on gradient descent (in contrast to essentially all practical GAN training) and relies on an oracle for minimizing the highly non-convex generator loss. This viewpoint is complementary to our approach. We establish results for learning the unknown distribution and analyze the commonly used gradient descent approach for learning GANs.

## 7   CONCLUSIONS

We haven taken a step towards a principled understanding of GAN dynamics. We define a simple yet rich model of GAN training and prove convergence of the corresponding dynamics. To the best of our knowledge, our work is the first to establish global convergence guarantees for a parametric GAN. We find an interesting dichotomy: If we take optimal discriminator steps, the training dynamics provably converge. In contrast, we show experimentally that the dynamics often fail if we take first order discriminator steps. We believe that our results provide new insights into GAN training and point towards a rich algorithmic landscape to be explored in order to further understand GAN dynamics.

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

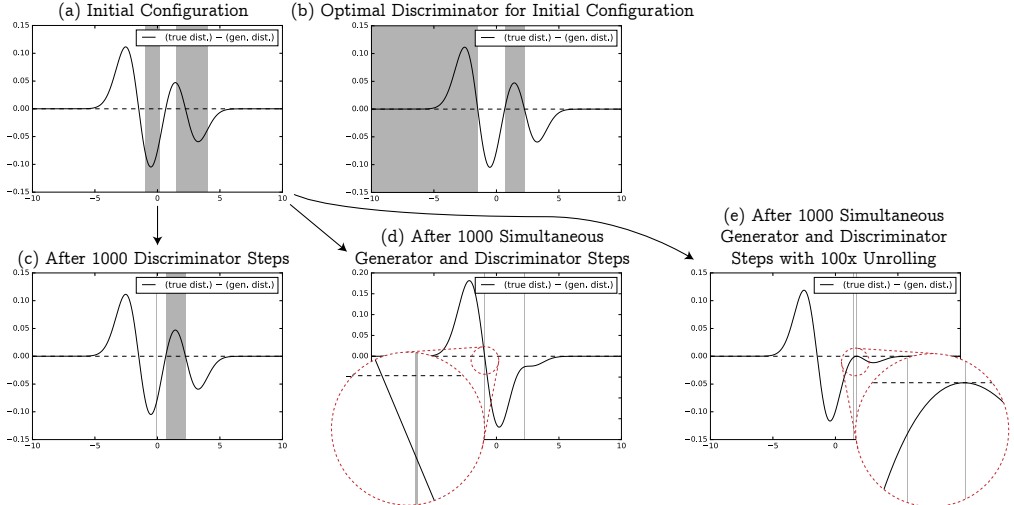

Figure 3: Example of Discriminator Collapse. The initial configuration has $\mu^* = \{-2, 2\}$, $\widehat{\mu} = \{-1, 2.5\}$, left discriminator $[-1, 0.2]$, and right discriminator $[-1, 2.5]$. The (multiplicative) step size used to generate (c), (d), and (e) was $0.3$.

## A    Discriminator Collapse: A Barrier for First Order Methods

As discussed above, our simple GAN dynamics are able to capture the same undesired behaviors that more sophisticated GANs exhibit. In addition to these behaviors, our dynamics enables us to discern another degenerate behavior which does not seem to have previously been observed in the literature. We call this behavior *discriminator collapse*.

We first explain this phenomenon using language specific to our GMM-GAN dynamics. In our dynamics, discriminator collapse occurs when a discriminator interval which originally had finite width is forced by the dynamics to have its width converge to $0$. This happens whenever this interval lies entirely in a region where the generator PDF is much larger than the discriminator PDF. We will shortly argue why this is undesirable.

In Figure 3, we show an example of discriminator collapse in our dynamics. Each plot in the figure shows the true PDF minus the PDF of the generators, where the regions covered by the discriminator are shaded. Plot (a) shows the initial configuration of our example. Notice that the leftmost discriminator interval lies entirely in a region for which the true PDF minus the generators' PDF is negative. Since the discriminator is incentivized to only have mass on regions where the difference is positive, the first order dynamics will cause the discriminator interval to collapse to have length zero if it is in a negative region. We see in Plot (c) that this discriminator collapses if we run many discriminator steps for this fixed generator. In particular, these steps do not converge to the globally optimal discriminator shown in Plot (b).

This collapse also occurs when we run the dynamics. In Plots (d) and (e), we see that after running the first order dynamics – or even unrolled dynamics – for many iterations, eventually *both* discriminators collapse. When a discriminator interval has length zero, it can never uncollapse, and moreover, its contribution to the gradient of the generator is zero. Thus these dynamics will never converge to the ground truth.

For general GANs, we view discriminator collapse as a situation when the local optimization landscape around the current discriminator encourages it to make updates which decrease its representational power. For instance, this could happen because the first order updates are unable to wholly follow the evolution of the optimal discriminator due to attraction of local maxima, and thus only capture part of the optimal discriminator's structure. We view understanding the exact nature of discriminator collapse in more general settings and interesting research problem to explore further.

# B   OMITTED DETAILS FROM SECTION 2

## B.1   TWO INTERVALS SUFFICE FOR $\mathcal{G}$

Here we formally prove (4). In fact, we will prove a slight generalization of this fact which will be useful later on.

We require the following theorem from Hummel and Gidas:

**Theorem B.1** ((Hummel & Gidas, 1984)). *Let $f$ be any analytic function with at most $n$ zeros. Then $f \circ \mathcal{N}(0, \sigma^2)$ has at most $n$ zeros.*

This allows us to prove:

**Theorem B.2.** *Any linear combination $F(x)$ of the probability density functions of $k$ Gaussians with the same variance has at most $k - 1$ zeros, provided at least two of the Gaussians have different means. In particular, for any $\mu \neq \nu$, the function $F(x) = D_\mu(x) - D_\nu(x)$ has at most 3 zeroes.*

*Proof.* If we have more than 1 Gaussian with the same mean, we can replace all Gaussians having that mean with an appropriate factor times a single Gaussian with that mean. Thus, we assume without loss of generality that all Gaussians have distinct means. We may also assume without loss of generality that all Gaussians have a nonzero coefficient in the definition of $F$.

Suppose the minimum distance between the means of any of the Gaussians is $\delta$. We first prove the statement when $\delta$ is sufficiently large compared to everything else. Consider any pair of Gaussians with consecutive means $\nu, \mu$. WLOG assume that $\mu > \nu = 0$. Suppose our pair of Gaussians has the same sign in the definition of $F$. In particular they are both strictly positive. For sufficiently large $\delta$, we can make the contribution of the other Gaussians to $F$ an arbitrarily small fraction of the whichever Gaussian in our pair is largest for all points on $[\nu, \mu]$. Thus, for $\delta$ sufficiently large, that there are no zeros on this interval.

Now suppose our pair of Gaussians have different signs in the definition of $F$. Without loss of generality, assume the sign of the Gaussian with mean $\nu$ is positive and the sign of the Gaussian with mean $\mu$ is negative. Then the PDF of the first Gaussian is strictly decreasing on $(\nu, \mu]$ and the PDF of the negation of the second Gaussian is decreasing on $[\nu, \mu)$. Thus, their sum is strictly decreasing on this interval. Similarly to before, by making $\delta$ sufficiently large, the magnitude of the contributions of the other Gaussians to the derivative in this region can be made an arbitrarily small fraction of the magnitude of whichever Gaussian in our pair contributes the most at each point in the interval. Thus, in this case, there is exactly one zero in the intervale $[\mu, \nu]$.

Also, note that there can be no zeros of $F$ outside of the convex hull of their means. This follows by essentially the same argument as the two positive Gaussians case above.

The general case (without assuming $\delta$ sufficiently large) follows by considering sufficiently skinny (nonzero variance) Gaussians with the same means as the Gaussians in the definition of $F$, rescaling the domain so that they are sufficiently far apart, applying this argument to this new function, unscaling the domain (which doesn't change the number of zeros), then convolving the function with an appropriate (very fat) Gaussian to obtain the real $F$, and invoking Theorem B.1 to say that the number of zeros does not increase from this convolution. $\square$

## B.2   THE FUNCTION $L$

In this section, we derive the form of $L$. By definition, we have

$$\sqrt{2\pi} L(\widehat{\mu}, \ell, r) = \sqrt{2\pi} \left( \mathbb{E}_{x \sim G_{\mu^*}}[D(x)] + \mathbb{E}_{x \sim G_\mu}[1 - D(x)] \right)$$
$$= \sqrt{2\pi} \left( \int_I G_{\mu^*}(x) - G_{\widehat{\mu}}(x)dx \right) + \sqrt{2\pi} \,,$$

where $I = [\ell_1, r_1] \cup [\ell_2, r_2]$. We then have

$$\sqrt{2\pi} L(\widehat{\mu}, \ell, r) = \sqrt{2\pi} \left( \sum_{i=1,2} \int_{\ell_i}^{r_i} G_{\mu^*}(x) - G_{\widehat{\mu}}(x) dx \right) + \sqrt{2\pi}$$

$$= \sum_{i=1,2} \sum_{j=1,2} \int_{\ell_i}^{r_i} e^{-(x-\mu_j^*)^2/2} - e^{-(x-\widehat{\mu}_j)^2/2} dx + \sqrt{2\pi} . \qquad (11)$$

It is not to hard to see from the Fundamental theorem of calculus that $L$ is indeed a smooth function of all parameters.

## C    ALTERNATIVE INDUCED DYNAMICS

Our focus in this paper is on the dynamics induced by, since it arises naturally from the form of the total variation distance  (3) and follows the canonical form of GAN dynamics (1). However, one could consider other equivalent definitions of total variation distance too. And these definitions could, in principle, induce qualitatively different behavior of the first order dynamics.

As mentioned in Section 3, an alternative dynamics could be induced by the definition of total variation distance given in (10). The corresponding loss function would be

$$L'(\mu, \ell, r) = |L(\mu, \ell, r)| = \left| \mathbb{E}_{x \sim G_{\mu^*}}[D(x)] + \mathbb{E}_{x \sim G_\mu}[1 - D(x)] \right| , \qquad (12)$$

i.e. the same as in (6) but with absolute values on the outside of the expression. Observe that this loss function does not actually fit the form of the general GAN dynamics presented in (1). However, it still constitutes a valid and fairly natural dynamics. Thus one could wonder whether similar behavior to the one we observe for the dynamics we actually study occurs also in this case.

To answer this question, we first observe that by the chain rule, the (sub)-gradient of $L'$ with respect to $\mu, \ell, r$ are given by

$$\nabla_\mu L'(\mu, \ell, r) = \mathrm{sgn}\left(L(\mu, \ell, r)\right) \nabla_\mu L(\mu, \ell, r)$$
$$\nabla_\ell L'(\mu, \ell, r) = \mathrm{sgn}\left(L(\mu, \ell, r)\right) \nabla_\ell L(\mu, \ell, r)$$
$$\nabla_r L'(\mu, \ell, r) = \mathrm{sgn}\left(L(\mu, \ell, r)\right) \nabla_r L(\mu, \ell, r) ,$$

that is, they are the same as for $L$ except modulated by the sign of $L$.

We now show that the *optimal* discriminator dynamics is identical to the one that we analyze in the paper (7), and hence still provably converge. This requires some thought; indeed a priori it is not even clear that the optimal discriminator dynamics are well-defined, since the optimal discriminator is no longer unique. This is because for any $\mu^*, \mu$, the sets $A_1 = \{x : G_{\mu^*}(x) \geq G_\mu(x)\}$ and $A_2 = \{x : G_\mu(x) \geq G_{\mu^*}(x)\}$ both achieve the maxima in  (10), since

$$\int_{A_1} G_\mu(x) - G_{\mu^*}(x) dx = - \int_{A_2} G_\mu(x) - G_{\mu^*}(x) dx . \qquad (13)$$

However, we show that the optimal discriminator dynamics are still well-formed. WLOG assume that $\int_{A_1} G_\mu(x) - G_{\mu^*}(x) dx \geq 0$, so that $A_1$ is also the optimal discriminator for the dynamics we consider in the paper. If we let $\ell^{(i)}, r^{(i)}$ be the left and right endpoints of the intervals in $A_i$ for $i = 1, 2$, we have that the update to $\mu$ induced by $(\ell^{(1)}, r^{(1)})$ is given by

$$\nabla_\mu L'(\mu, \ell^{(1)}, r^{(1)}) = \nabla_\mu L(\mu, \ell^{(1)}, r^{(1)}) ,$$

so the update induced by $(\ell^{(1)}, r^{(1)})$ is the same as the one induced by the optimal discriminator dynamics in the paper. Moreover, the update to $\mu$ induced by $(\ell^{(2)}, r^{(2)})$ is given by

$$\nabla_\mu L'(\mu, \ell^{(2)}, r^{(2)}) = \mathrm{sgn}\left( L(\mu, \ell^{(2)}, r^{(2)}) \right) \nabla_\mu L(\mu, \ell^{(2)}, r^{(2)})$$

$$\overset{(a)}{=} -\nabla_\mu(-L(\mu, \ell^{(1)}, r^{(1)}))$$

$$= \nabla_\mu L(\mu, \ell^{(1)}, r^{(1)}) ,$$

where (a) follows from the assumption that $\int_{A_1} G_\mu(x) - G_{\mu^*}(x)dx \geq 0$ and from (13), so it is also equal to the the the one induced by the optimal discriminator dynamics in the paper. Hence the optimal discriminator dynamics are well-formed and unchanged from the optimal discriminator dynamics described in the paper.

Thus the question is whether the first order approximation of this dynamics and/or the unrolled first order dynamics exhibit the same qualitative behavior too. To evaluate the effectiveness, we performed for these dynamics experiments analogous to the ones summarized in Figure 2 in the case of the dynamics we actually analyzed. The results of these experiments are presented in Figure 4. Although the probability of success for these dynamics is higher, they still often do not converge. We can thus see that a similar dichotomy occurs here as in the context of the dynamics we actually study. In particular, we still observe the discriminator collapse phenomena in these first order dynamics.

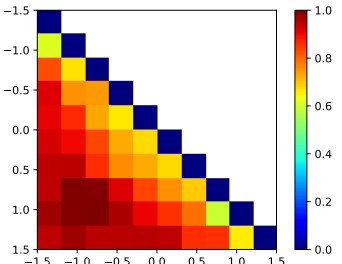 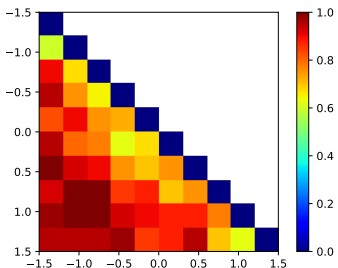

Figure 4: Heatmap of success probability for random discriminator initialization for regular GAN training, unrolled GAN training with dynamics induced by 12

### C.1 WHY DOES DISCRIMINATOR COLLAPE STILL HAPPEN?

It might be somewhat surprising that even with absolute values discriminator collapse occurs. Originally the discriminator collapse occurred because if an interval was stuck in a negative region, it always subtracts from the value of the loss function, and so the discriminator is incentivized to make it disappear. Now, since the value of the loss is always nonnegative, it is not so clear that this still happens.

Despite this, we still observe discriminator collapse with these dynamics. Here we describe one simple scenario in which discriminator collapse still occurs. Suppose the discriminator intervals have left and right endpoints $\ell, r$ and $L(\mu, \ell, r) > 0$. Then, if it is the case that $\int_{\ell_i}^{r_i} G_{\mu^*}(x) - G_\mu(x)dx < 0$ for some $i = 1, 2$. that is, on one of the discriminator intervals the value of the loss is negative, then the discriminator is still incentivized locally to reduce this interval to zero, as doing so increases both $L(\mu, \ell, r)$ and hence $L'(\mu, \ell, r)$. Symmetrically if $L(\mu, \ell, r) < 0$ and there is a discriminator interval on which the loss is positive, the discriminator is incentivized locally to reduce this interval to zero, since that increases $L'(\mu, \ell, r)$. This causes the discriminator collapse and subsequently causes the training to fail to converge.

## D OMITTED PROOFS FROM SECTION 4.1

This appendix is dedicated to a proof of Theorem 4.1. We start with some remarks on the proof techniques for these main lemmas. At a high level, Lemmas 4.3, 4.5, 4.6 all follow from involved case analyses. Specifically, we are able to deduce structure about the possible discriminator intervals by reasoning about the structure of the current mean estimate $\widehat{\mu}$ and the true means. From there we are able to derive bounds on how these discriminator intervals affect the derivatives and hence the update functions.

To prove Lemma 4.4, we carefully study the evolution of the optimal discriminator as we make small changes to the generator. The key idea is to show that when the generator means are far from the true means, then the zero crossings of $F(\widehat{\mu}, x)$ cannot evolve too unpredictably as we change $\widehat{\mu}$. We do so by showing that locally, in this setting $F$ can be approximated by a low degree polynomial

with large coefficients, via bounding the condition number of a certain Hermite Vandermonde matrix. This gives us sufficient control over the local behavior of zeros to deduce the desired claim. By being sufficiently careful with the bounds, we are then able to go from this to the full generality of the lemma. We defer further details to Appendix D.

## D.1    SETUP

By inspection on the form of (11), we see that the gradient of the function $f_{\mu^*}(\widehat{\mu})$ if it is defined must be given by

$$\frac{\partial f_{\mu^*}}{\partial \widehat{\mu}_i} = \frac{1}{\sqrt{2\pi}} \sum_{i=1,2} \left( e^{-(\widehat{\mu}_i - r_i)^2/2} - e^{-(\widehat{\mu}_i - \ell_i)^2/2} \right) \ .$$

Here $I_i = [\ell_i, r_i]$ are the intervals which achieve the supremum in (4). While these intervals may not be unique, it is not hard to show that this value is well-defined, as long as $\widehat{\mu} \neq \mu^*$, that is, when the optimal discriminator intervals are unique as sets.

Recall $F_{\mu^*}(\widehat{\mu}, x) = G_{\mu^*}(x) - G_{\widehat{\mu}}(x)$.

## D.2    BASIC MATH FACTS

Before we begin, we require the following facts.

We first need that the Gaussian, and any fixed number of derivatives of the Gaussian, are Lipschitz functions.

**Fact D.1.** *For any constant $i$, there exists a constant $B$ such that for all $x, \mu \in \mathbb{R}$, $\frac{d^i}{dx^i} \mathcal{N}(x, \mu, \sigma^2 = 1) \leq B$.*

*Proof.* Note that every derivative of the Gaussian PDF (including the $0th$) is a bounded function. Furthermore, all these derivatives eventually tend to $0$ whenever the input goes towards $\infty$ or $-\infty$. Thus, any particular derivative is bounded by a constant for all $\mathbb{R}$. Furthermore, shifting the mean of the Gaussian does not change the set of values the derivatives of its derivative takes (only their locations). $\qquad\square$

We also need the following bound on the TV distance between two Gaussians, which is folklore, and is easily proven via Pinsker's inequality.

**Fact D.2** (folklore). *If two univariate Gaussians with unit variance have means within distance at most $\Delta$ then their TV distance is at most $O(1) \cdot \Delta$.*

This immediately implies the following, which states that $f_{\mu^*}$ is Lipschitz.

**Corollary D.3.** *There is some absolute constant $C$ so that for any $\mu, \nu$, we have $|f_{\mu^*}(\mu) - f_{\mu^*}(\nu)| \leq C\|\mu - \nu\|_2$.*

We also need the following basic analysis fact:

**Fact D.4** (folklore). *Suppose $g : \mathbb{R}^d \to \mathbb{R}$ is $B$-Lipschitz for some $B$. Then $g$ is differentiable almost everywhere.*

This implies that $f_{\mu^*}$ is indeed differentiable except on a set of measure zero. As mentioned previously, we will always assume that we never land within this set during our analysis.

## D.3    PROOF OF THEOREM 4.1 GIVEN LEMMATA

Before we prove the various lemmata described in the body, we show how Theorem 4.1 follows from them.

*Proof of Theorem 4.1.* Set $\delta'$ be a sufficiently small constant multiple of $\delta$. Provided we make the nonzero constant factor on the step size sufficiently small (compared to $\delta'/\delta$), and the exponent on $\delta$ in the magnitude step size at least one, the magnitude of our step size will be at most $\delta'$. Thus, in any

step where $\widehat{\mu} \in \mathrm{Opt}(\delta')$, we end the step outside of this set but still in $\mathrm{Opt}(2\delta')$. By Lemma D.2, for a sufficiently small choice of constant in the definition of $\delta'$, the TV-distance at the end of such a step will be at most $\delta$.

Contrapositively, in any step where the TV-distance at the start is more than $\delta$, we will have at the start that $\widehat{\mu} \notin \mathrm{Opt}(\delta')$. Then, it suffices to prove that the step decreases the total variation distance additively by at least $1/\operatorname{poly}(C, e^{C^2}, 1/\delta)$ in this case. For appropriate choices of constants in expression for the step size (sufficiently small multiplicative and sufficiently large in the exponent), this is immediately implied by Lemma 4.4 and Lemma 4.2 provided that $\mu^*, \widehat{\mu}, \widehat{\mu}' \in B(2C)$ and $|\widehat{\mu}_1 - \widehat{\mu}_2| \geq \delta$ at the beginning of each step. The condition that we always are within $B(2C)$ at the start of each step is proven in Lemma 4.6 and the condition that the means are separated (ie., that we don't have mode collapse) is proven in Lemma 4.5. □

It is interesting that a critical component of the above proof involves proving explicitly that mode collapse does not occur. This suggests the possibility that understanding mode collapse may be helpful in understanding convergence of Generative Adversarial Models and Networks.

### D.4 PROOF OF LEMMA 4.3

In this section we prove Lemma 4.3. We first require the following fact:

**Fact D.5** ((Markov, 1892)). *Let $p(x) = \sum_{i=0}^{d} c_j x^j$ be a degree $d$ polynomial so that $|p(x)| \leq 1$ for all $x \in [-1, 1]$. Then $\max_{0 \leq j \leq d} |c_j| \leq (\sqrt{2}+1)^d$. More generally, if $|p(x)| \leq \alpha$ for all $x \in [-\rho, \rho]$, then $\max_{0 \leq j \leq d} |c_j \rho^j| \leq O(\alpha)$.*

We also have the following, elementary lemma:

**Lemma D.6.** *Suppose $\widehat{\mu}_2 > \mu_i^*$ for all $i$. Then there is some $x > \widehat{\mu}_2$ so that $F_{\mu^*}(\widehat{\mu}, x) < 0$.*

We are now ready to prove Lemma 4.3

*Proof of Lemma 4.3.* We proceed by case analysis on the arrangement of the $\widehat{\mu}$ and $\mu^*$.

**Case 1:** $\mu_1^* < \widehat{\mu}_1$ **and** $\mu_2^* < \widehat{\mu}_2$ In this case we have $F_{\mu^*}(\widehat{\mu}, x) \leq 0$ for all $x \geq \widehat{\mu}_2$. Hence the optimal discriminators are both to the left of $\widehat{\mu}_2$. Moreover, by a symmetric logic we have $F_{\mu^*}(\widehat{\mu}, x) \geq 0$ for all $x \leq \mu_1^*$, so the optimal discriminator has an interval of the form $I_1 = [-\infty, r_1]$ and possibly $I_2 = [l_2, r_2]$ where $r_1 < l_2 < r_2 < \widehat{\mu}_2$. Then it is easy to see that $\frac{\partial f}{\partial \widehat{\mu}_2}(\widehat{\mu}_2) \geq \frac{1}{\sqrt{2\pi}} e^{-(\widehat{\mu}_2 - r_2)^2/2} \geq \frac{1}{\sqrt{2\pi}} e^{-2C^2}$.

**Case 2:** $\widehat{\mu}_1 < \mu_1^*$ **and** $\widehat{\mu}_2 < \mu_2^*$ This case is symmetric to Case (1).

**Case 3:** $\widehat{\mu}_1 < \mu_1^* < \mu_2^* < \widehat{\mu}_2$ By Lemma D.6, we know that $F_{\mu^*}(\widehat{\mu}, x) < 0$ for some $x \geq \widehat{\mu}_2$, and similarly $F_{\mu^*}(\widehat{\mu}, x) < 0$ for some $x \leq \widehat{\mu}_1$. Since clearly $F(\mu^*)(\widehat{\mu}, x) > 0$ for $x \in [\mu_1^*, \mu_2^*]$, by Theorem B.2 and continuity, the optimal discriminator has one interval. Denote it by $I = [\ell, r]$, so that we have $\ell \leq \mu_1^*$ and $r \geq \mu_2^*$. Suppose $\ell \geq \widehat{\mu}_1$. Then

$$\frac{\partial f}{\partial \widehat{\mu}_1}(\widehat{\mu}_1) = \frac{1}{\sqrt{2\pi}} \left( e^{-(\widehat{\mu}_1 - \ell)^2/2} - e^{-(\widehat{\mu}_1 - r)^2/2} \right)$$

$$= \frac{1}{\sqrt{2\pi}} e^{-(\widehat{\mu}_1 - \ell)^2/2} \left( 1 - e^{-(\widehat{\mu}_1 - \ell)(r - \ell)} e^{-(r - \ell)^2/2} \right)$$

$$\geq \frac{1}{\sqrt{2\pi}} e^{-2C^2} \left( 1 - e^{-\delta^2/2} \right) .$$

We get the symmetric bound on $\frac{\partial f_{\mu^*}}{\partial \widehat{\mu}_2}(\widehat{\mu}_2)$ if $r \leq \widehat{\mu}_2$. The final case is if $\ell < \widehat{\mu}_1 < \widehat{\mu}_2 < r$. Consider the auxiliary function

$$H(\mu) = e^{-(\ell - \mu)^2/2} - e^{-(r - \mu)^2/2} .$$

On the domain $[\ell, r]$, this function is monotone decreasing. Moreover, for any $\mu \in [\ell, r]$, we have

$$
\begin{aligned}
H'(\mu) &= (\ell - \mu)e^{-(\ell-\mu)^2/2} - (r - \mu)e^{-(r-\mu)^2/2} \\
&\leq -\frac{r - \ell}{2}e^{-(r-\ell)^2/8} \\
&\leq -\frac{\gamma}{2}e^{-\gamma^2/8} \ .
\end{aligned}
$$

In particular, this implies that $H(\widehat{\mu}_1) < H(\widehat{\mu}_2) - \gamma^2 e^{-\gamma^2/8}/2$, so at least one of $H(\widehat{\mu}_2)$ or $H(\widehat{\mu}_1)$ must be $\gamma^2 e^{-\gamma^2/8}/4$ in absolute value. Since $\frac{\partial f_{\mu^*}}{\partial \widehat{\mu}_i}(\widehat{\mu}_i) = H(\widehat{\mu}_i)$, this completes the proof in this case.

**Case 4:** $\mu_1^* < \widehat{\mu}_1 < \widehat{\mu}_2 < \mu_2^*$ By a symmetric argument to Case 3, we know that the optimal discriminator intervals are of the form $(-\infty, r]$ and $[\ell, \infty)$ for some $r < \widehat{\mu}_1 < \widehat{\mu}_2 < \ell$. The form of the derivative is then exactly the same as in the last sub-case in Case 3 with signs reversed, so the same bound holds here.

$\square$

### D.5 PROOF OF LEMMA 4.4

We now seek to prove Lemma 4.4. Before we do so, we need to get lower bounds on derivatives of finite sums of Gaussians with the same variance. In particular, we first show:

**Lemma D.7.** *Fix $\gamma \geq \delta > 0$ and $C \geq 1$. Suppose we have $\mu^*, \widehat{\mu} \in B(C)$, $\mu^*, \widehat{\mu} \in \text{Sep}(\gamma)$, with $\widehat{\mu} \notin \text{Rect}(\delta)$, where all these vectors have constant length $k$. Then, for any $x \in [-C, C]$, we have that $|\frac{d^i}{dx^i}F_{\mu^*}(\widehat{\mu}, x)| \geq \Omega(1) \cdot (\delta/C)^{O(1)}e^{-C^2/2}$ for some $i = 0, \ldots, 2k - 1$.*

*Proof.* Observe that the value of the $i$th derivative of $F_{\mu^*}(\widehat{\mu}, x)$ for any $x$ is given by

$$
\frac{d^i}{dx^i}F_{\mu^*}(\widehat{\mu}, x) = \frac{1}{\sqrt{2\pi}}\sum_{j=1}^{2k} w_j(-1)^i H_i(z_j)e^{-z_j^2/2} \ ,
$$

where $w_j \in \{-1/k, 1/k\}$, the $z_j$ is either $x - \mu_j^*$ or $x - \widehat{\mu}_j$, and $H_i(z)$ is the $i$th (probabilist's) Hermite polynomial. Note that the $(-1)^i H_i$ are orthogonal with respect to the Gaussian measure over $\mathbb{R}$, and are orthonormal after some finite scaling that depends only on $i$ and is therefore constant. Hence, if we form the matrix $M_{ij} = (-1)^i H_i(x - z_j)$, if we define $u_i = \frac{d^i}{dx^i}F_{\mu^*}(\widehat{\mu}, x)$ for $i = 0, \ldots, 2k - 1$, we have that $Mv = u$, where $v_j = \frac{1}{\sqrt{2\pi}}w_j e^{-(x-z_j)^2/2}$. By assumption, we have $\|v\|_2 \geq \Omega(\sqrt{k} \cdot e^{-C^2/2}) = \Omega(e^{-C^2/2})$. Thus, to show that some $u_i$ cannot be too small, it suffices to show a lower bound on the smallest singular value of $M$. To do so, we leverage the following fact, implicit in the arguments of (Gautschi, 1990):

**Theorem D.8** ((Gautschi, 1990)). *Let $p_r(z)$ be family of orthonormal polynomials with respect to a positive measure $d\sigma$ on the real line for $r = 1, 2, \ldots, t$ and let $z_1, \ldots, z_t$ be arbitrary real numbers with $z_i \neq z_j$ for $i \neq j$. Define the matrix $V$ given by $V_{ij} = p_i(z_j)$. Then, the smallest singular value of $V$, denoted $\sigma_t(V)$, is at least*

$$
\sigma_{min}(V) \geq \left(\int_{\mathbb{R}}\sum_{r=1}^{t}\ell_r(y)^2 d\sigma(y)\right)^{-1/2} \ ,
$$

*where $\ell_r(y) = \prod_{s \neq r}\frac{y - z_s}{z_r - z_s}$ is the Langrange interpolating polynomial for the $z_r$.*

Set $p_r = H_{r-1}$ $t = 2k$, and $\sigma$ as the Gaussian measure; then apply the theorem. Observe that for any $i, j$, we have $|z_i - z_j| \geq \min(\delta, \gamma) \geq \delta$ and $|z_i| \leq C$. Hence the largest coefficient of any Lagrange interpolating polynomial through the $z_i$ is at most $(\frac{C}{\delta})^{2k-1}$ with degree $2k - 1$. So, the square of

any such polynomial has degree at most $2(2k-1)$ and max coefficient at most $2k(\frac{C}{\delta})^{2(2k-1)}$ This implies that

$$
\begin{aligned}
\int_{\mathbb{R}} \sum_{r=1}^{2k} \ell_r(y)^2 \, \mathrm{d}\sigma(y) &= \sum_{r=1}^{2k} \int_{\mathbb{R}} \ell_r(y)^2 \, \mathrm{d}\sigma(y) \\
&\leq \sum_{r=1}^{2k} 2(2k-1) \cdot 2k \left(\frac{C}{\delta}\right)^{2(2k-1)} \max_{s \in [2(2k-1)]} \int_{\mathbb{R}} y^s \mathrm{d}\sigma(y) \\
&\leq O(1) \cdot \left(\frac{C}{\delta}\right)^{4k} \max_{s \in [4k]} \int_{-\infty}^{\infty} y^s e^{-y^2/2} \, \mathrm{d}y \\
&\leq O(1) \cdot \left(\frac{C}{\delta}\right)^{O(1)}.
\end{aligned}
$$

Hence by Theorem D.8 we have that $\sigma_{\min}(V) \geq \Omega(1) \cdot \left(\frac{\delta}{C}\right)^{O(1)}$. Therefore, we have that $\|u\|_2 \geq \Omega(1) \cdot (\delta/C)^{O(1)} e^{-C^2/2}$, which immediately implies the desired statement. $\qquad\square$

We next show that the above Lemma can be slightly generalized, so that we can replace the condition $\widehat{\mu} \notin \mathrm{Rect}(\delta)$ with $\widehat{\mu} \notin \mathrm{Opt}(\delta)$.

**Lemma D.9.** *Fix $C \geq 1 \geq \gamma \geq \delta \geq \Xi > 0$. Suppose we have $\mu^*, \widehat{\mu} \in B(C)$, $\mu^*, \widehat{\mu} \in \mathrm{Sep}(\gamma)$, with $\widehat{\mu} \notin \mathrm{Opt}(\delta)$. Then for any $x \in [-C, C]$, we have that $|\frac{d^i}{dx^i} F_{\mu^*}(\widehat{\mu}, x)| \geq \Omega(1) \cdot (\delta e^{-C^2}/C)^{O(1)}$ for some $i = 0, \ldots, 3$.*

*Proof.* Let $\Xi$ be of the form $\Omega(1) \cdot (\delta e^{-C^2}/C)^{O(1)}$, where we will pick its precise value later. Lemma D.7 with $\delta$ in that Lemma set to $\Xi$ and $k = 2$ proves the special case when $\widehat{\mu} \notin \mathrm{Rect}(\Xi)$. Thus, the only remaining case is when $\widehat{\mu}_i$ is close to $\mu_i^*$ for some $i$ and far away for the other $i$. Without loss of generality, we assume the first entries are the close pair. Then we have $|\widehat{\mu}_1 - \mu_1^*| \leq \Xi$ and $|\widehat{\mu}_2 - \mu_2^*| \geq \delta$.

There are four terms in the expression for $\frac{d^i}{dx^i} F_{\mu^*}(\widehat{\mu}, x)$ corresponding to each of $\widehat{\mu}_1, \widehat{\mu}_2, \mu_1^*, \mu_2^*$. Lemma D.7 with $\delta = \Xi$ and $k = 1$ implies that the contribution of the $\widehat{\mu}_2$ and $\mu_2^*$ terms to at least one of the 0th through 3rd derivatives has magnitude at least $\Omega(1) \cdot (\delta e^{-C^2}/C)^{O(1)}$. Fact D.2 and Lemma D.10 (below) imply that the contribution of the $\widehat{\mu}_1$ and $\mu_1^*$ terms to these derivatives has magnitude at most $O(1) \cdot \Xi^4$. Thus, there exists a $\Xi = \Omega(1) \cdot (\delta/C)^{O(1)} e^{-C^2/2}$ such that the magnitude of the contribution of these second two terms is less than half that of the first two, which gives a lower bound on the magnitude of the sum of all the terms of $\Omega(1) \cdot (\delta/C)^{O(1)} e^{-C^2/2}$. $\qquad\square$

We now show that any function which always has at least one large enough derivative—including its 0th derivaive—on some large enough interval must have a nontrivial amount of mass on the interval.

**Lemma D.10.** *Let $0 < \xi < 1$ and $t \in \mathbb{N}$. Let $F(x) : \mathbb{R} \to \mathbb{R}$ be a $(t+1)$-times differentiable function such that at every point $x$ on some interval $I$ of length $|I| \geq \xi$, $F(x) \geq 0$ and there exists an $i = i(x) \in 0, \ldots, t$ such that $|\frac{d^i}{dx^i} F(x)| \geq B'$ for some $B'$. Also suppose $|\frac{d^{t+1}}{dx^{t+1}} F(x)| \leq B$ for some $B$. Then,*

$$
\int_z^y F(x) dx \geq \left( \frac{B' \cdot (\Omega(1) \cdot \xi)^{t+1} \cdot \min[(B'/B)^{t+2}, 1]}{(t+1)! \cdot (t+1)} \right).
$$

*Proof.* Let $0 < a < 1$ be a non-constant whose value we will choose later. If $I$ has length more than $a\xi$, truncate it to have this length. Let $z$ denote the midpoint of $I$. By assumption, we know that there is some $i \in 0, \ldots, t$ such that $|\frac{d^i}{dx^i} F(x)| > \xi$. Thus, by Taylor's theorem, we have that $F(\widehat{\mu}, x) \geq p(x - z) - (B/(t+1)!) \cdot |x - z|^{t+1}$ for some degree $t$ polynomial $p$ that has some coefficient of magnitude at least $B'/t!$.

Thus, if we let $G(y) = \int_z^y p(x) dx$, then $G(y)$ is a degree $t + 1$ polynomial with some coefficient which is at least $B'/(t! \cdot t)$. By the nonnegativity of $F$ on $I$, we have that $G$ is nonnegative on

$[-a\xi/2, a\xi/2]$. By this and the contrapositive of Fact D.5 (invoked with $\alpha$ set to a sufficiently small nonzero constant multiple of $B$), we have for some such $y$ and some constant $B'' > 0$ that $G(y) = |G(y)| \geq B''(|I|/2)^{t+1}B'/(t! \cdot t)$. Therefore, at this point, we have

$$\int_z^y F(x)dx \geq G(y) - \int_z^y (B/(t+1)!) \cdot |x - z|^{t+1}\mathrm{d}x$$

$$\geq \frac{B''a^{t+1}(\xi/2)^{t+1}B'}{t! \cdot t} - \frac{B(a\xi/2)^{t+2}}{(t+1)! \cdot (t+1)}$$

$$\geq \left( \frac{a^{t+1}(\xi/2)^{t+1}(B''B' - B\xi a/2)}{(t+1)! \cdot (t+1)} \right)$$

$$\geq \left( \frac{a^{t+1}(\xi/2)^{t+1}(B''B' - Ba/2)}{(t+1)! \cdot (t+1)} \right).$$

If $B'B'' \leq B$, we set $a = B'B''/B \leq 1$ which gives

$$\int_z^y F(x)dx \geq \left( \frac{(B')^{t+2}(\Omega(1) \cdot \xi/B)^{t+1}}{(t+1)! \cdot (t+1)} \right).$$

Otherwise, $B'B'' \geq B$ and we perform this substitution along with $a = 1$ which gives the similar bound

$$\int_z^y F(x)dx \geq \left( \frac{B'(\Omega(1) \cdot \xi)^{t+1}}{(t+1)! \cdot (t+1)} \right).$$

Together, these bounds imply that we always have

$$\int_z^y F(x)dx \geq \left( \frac{B' \cdot (\Omega(1) \cdot \xi)^{t+1} \cdot \min[(B'/B)^{t+2}, 1]}{(t+1)! \cdot (t+1)} \right).$$

$\square$

This allows us to prove the following lemma, which lower bounds how much mass $F$ can put on any interval which is moderately large. Formally:

**Lemma D.11.** *Fix $C \geq 1 \geq \gamma \geq \delta > 0$. Let $K = \Omega(1) \cdot (\delta e^{-C^2}/C)^{O(1)}$ be the $K$ for which Lemma 4.3 is always true with those parameters. Let $\mu^*, \widehat{\mu}$ be so that $\widehat{\mu} \notin \mathrm{Opt}(\delta)$, $\widehat{\mu}, \mu^* \in \mathrm{Sep}(\gamma)$, and $\mu^*, \widehat{\mu} \in B(C)$. Then, there is a $\xi = \Omega(1) \cdot (\delta/C)^{O(1)}e^{-C^2})^{O(1)}$ such that for any interval $I$ of length $|I| \geq \xi$ which satisfies $I \cap [-C - 2\sqrt{\log(100/K)}, C + 2\sqrt{\log(100/K)}] \neq \varnothing$ and on which $F(\widehat{\mu}, x)$ is nonnegative, we have*

$$\int_I |F(\widehat{\mu}, x)|dx \geq \Omega(1) \cdot (\delta e^{-C^2}/C)^{O(1)}\xi^{O(1)}.$$

*Proof.* By Lemma D.9 with $C$ in that lemma set to $C + 2\sqrt{\log(100/K)}$, we get a lower bound of $\Omega(1) \cdot (\delta e^{-C^2}/C)^{O(1)}$ on the magnitude of at least one of the 0th through 3rd derivatives of $F(\widehat{\mu}, x)$ with respect to $x$. Set $\xi$ equal to a sufficiently small (nonzero) constant times this value.

By Fact D.1 there exists a constant $B$ such that the magnitude of the fifth derivative of $F(\widehat{\mu}, x)$ with respect to $x$—which is a linear combination of four fifth derivatives of Gaussians with constant coefficients—is at most $B$.

By Lemma D.10 applied to $F(\widehat{\mu}, x)$ as a function of $x$, we have $\int_I F(\widehat{\mu}, x)dx \geq \Omega(1) \cdot \xi^6$. $\square$

Now we can prove Lemma 4.4.

*Proof of Lemma 4.4.* Let $A = [C - 2\sqrt{\log(100/K)}, C + 2\sqrt{\log(100/K)}]$ where $K = \Omega(1) \cdot (\delta e^{-C^2}/C)^{O(1)}$ is the $K$ for which Lemma 4.3 is always true with those parameters.

Let $Z^\pm$ denote the set of all $x \in A$ for which $F(\widehat{\mu}', x)$ and $F(\widehat{\mu}, x)$ have different nonzero signs. Let $Z^+$ denote the subset of $Z^\pm$ where $F(\widehat{\mu}', x) > 0 > F(\widehat{\mu}, x)$ and $Z^-$ denote the subset where $F(\widehat{\mu}', x) < 0 < \widehat{\mu}, x)$. Then $Z^\pm = Z^+ \cup Z^-$ and $Z^+, Z^-$ are disjoint and Lebesgue-measurable. If $\text{vol}(Z^+) \le \text{vol}(Z^-)$, switch $\widehat{\mu}$ and $\widehat{\mu}'$ so that $\text{vol}(Z^+) \ge \text{vol}(Z^-)$.

Note that $Z^+$ can be obtained by making cuts in the real line at the zeros of $F(\widehat{\mu}', x)$, $F(\widehat{\mu}, x)$, and $F(\widehat{\mu}', x) - F(\widehat{\mu}, x)$, then taking the union of some subset of the open intervals induced by these cuts. By Theorem B.2, the total number of such intervals is $O(1)$. Thus, $Z^+$ is the union of a constant number of open intervals. By similar arguments, $Z^-$ is also the union of a constant number of open intervals.

We now prove that $\text{vol}(Z^+), Z^{-1} \le O(1) \cdot \|\widehat{\mu}' - \widehat{\mu}\|_1^{\Theta(1)} \cdot (\delta e^{-C^2}/C)^{-O(1)}$. Since $\text{vol}(Z^+) \ge \text{vol}(Z^-)$, it suffices to prove $\text{vol}(Z^+) \le O(1) \cdot \|\widehat{\mu}' - \widehat{\mu}\|_1^{\Theta(1)} \cdot (\delta e^{-C^2}/C)^{-O(1)}$. Note also that by Lemma D.2, each of these intervals has mass under $F(\widehat{\mu}', x)$ at most $\int_{\mathbb{R}} |F(\widehat{\mu}', x) - F(\widehat{\mu}, x)| dx \le O(1) \cdot \|\widehat{\mu}' - \widehat{\mu}\|_1$. By Lemma D.11 and Lemma 4.3, each of these intervals has length at most $O(1) \cdot \|\widehat{\mu}' - \widehat{\mu}\|_1^{\Theta(1)} \cdot (\delta e^{-C^2}/C)^{-O(1)}$. Since there are at most a constant number of such intervals, this is also a bound on $\text{vol}(Z^+)$ (and $\text{vol}(Z^-)$).

Let $Y$ denote the set of $x \in A$ on which both $F(\widehat{\mu}, x)$ and $F(\widehat{\mu}', x)$ are nonnegative. Let $X, X'$ denote the $x \notin A$ for which $F(\widehat{\mu}, x)$ and $F(\widehat{\mu}', x)$ are respectively positive. Let $W, W'$ denote, respectively, the sets of endpoints of the union of the optimal discriminators for $\widehat{\mu}, \widehat{\mu}'$. Then the union of the optimal discriminators for $\widehat{\mu}, \widehat{\mu}'$ are respectively $Y \cup Z^- \cup X \cup W$ and $Y \cup Z^+ \cup X' \cup W'$. Furthermore, each of these two unions is given by some constant number of closed intervals and more specifically, that $X, X'$ each contain at most two intervals by Lemma B.2. Thus, we have for any $i$ that

$$
\left| \frac{\partial}{\partial \widehat{\mu}_i} \text{TV}(\mu^*, \widehat{\mu}) \Big|_{\widehat{\mu}}^{\widehat{\mu}'} \right|
$$

$$
= \left| \int_{Y \cup Z^+ \cup W' \cup X'} \frac{\mathrm{d}}{\mathrm{d}x} e^{(x - \widehat{\mu}_i')^2/2} \mathrm{d}x - \int_{Y \cup Z^- \cup W \cup X} \frac{\mathrm{d}}{\mathrm{d}x} e^{(x - \widehat{\mu}_i)^2/2} \mathrm{d}x \right|
$$

$$
\le \left| \int_{Y \cup Z^+ \cup W' \cup X'} \frac{\mathrm{d}}{\mathrm{d}x} e^{(x - \widehat{\mu}_i)^2/2} \mathrm{d}x - \int_{Y \cup Z^- \cup W \cup X} \frac{\mathrm{d}}{\mathrm{d}x} e^{(x - \widehat{\mu}_i)^2/2} \mathrm{d}x \right|
$$

$$
+ O(1) \cdot |\widehat{\mu}_i' - \widehat{\mu}_i|,
$$

by Lipschitzness, and so

$$
\left| \frac{\partial}{\partial \widehat{\mu}_i} \mathrm{TV}(\mu^*, \widehat{\mu}) \Big|_{\widehat{\mu}}^{\widehat{\mu}'} \right|
$$

$$
= \left| \int_{Z^+ \cup X'} \frac{\mathrm{d}}{\mathrm{d}x} e^{(x-\widehat{\mu}_i)^2/2} \mathrm{d}x - \int_{Z^- \cup X} \frac{\mathrm{d}}{\mathrm{d}x} e^{(x-\widehat{\mu}_i)^2/2} \mathrm{d}x \right|
$$

$$
+ O(1) \cdot |\widehat{\mu}_i' - \widehat{\mu}_i|
$$

$$
\leq \left| \int_{Z^+} \frac{\mathrm{d}}{\mathrm{d}x} e^{(x-\widehat{\mu}_i)^2/2} \mathrm{d}x \pm \frac{4}{100} \cdot K - \int_{Z^-} \frac{\mathrm{d}}{\mathrm{d}x} e^{(x-\widehat{\mu}_i)^2/2} \mathrm{d}x \pm \frac{4}{100} \cdot K \right|
$$

$$
+ O(1) \cdot |\widehat{\mu}_i' - \widehat{\mu}_i|
$$

$$
\leq \left| \int_{Z^+} \frac{\mathrm{d}}{\mathrm{d}x} e^{(x-\widehat{\mu}_i)^2/2} \mathrm{d}x \right| + \left| \int_{Z^-} \frac{\mathrm{d}}{\mathrm{d}x} e^{(x-\widehat{\mu}_i)^2/2} \mathrm{d}x \right|
$$

$$
+ \frac{8}{100} \cdot K + O(1) \cdot |\widehat{\mu}_i' - \widehat{\mu}_i|
$$

$$
\leq 2\mathrm{vol}(Z^+) \left| \sup_{x \in \mathbb{R}} \frac{\mathrm{d}}{\mathrm{d}x} e^{(x-\widehat{\mu}_i)^2/2} \right| + \frac{8}{100} \cdot K + O(1) \cdot |\widehat{\mu}_i' - \widehat{\mu}_i|
$$

$$
\leq O(1) \cdot \|\widehat{\mu}' - \widehat{\mu}\|_2^{\Theta(1)} \cdot (\delta e^{-C^2}/C)^{-O(1)} + \frac{8}{100} \cdot K .
$$

This bound also upper bounds $\|\nabla f_{\mu^*}(\widehat{\mu}') - \nabla f_{\mu^*}(\widehat{\mu})\|_2$ up to a constant factor. Thus, if we choose our step to have magnitude $\|\widehat{\mu}' - \widehat{\mu}\|_2 \leq \Omega(1) \cdot (\delta e^{-C^2}/C)^{O(1)}$ with appropriate choices of constants, we get

$$
\|\nabla f_{\mu^*}(\widehat{\mu}') - \nabla f_{\mu^*}(\widehat{\mu})\|_2 \leq K/2 \leq \|\nabla f_{\mu^*}(\widehat{\mu})\|_2/2 ,
$$

as claimed $\qquad\qquad\square$

### D.6   PROOF OF LEMMA 4.5

We now prove Lemma 3.4, which forbids mode collapse.

*Proof of Lemma 4.5.* Since $\eta \leq \delta$, if $|\widehat{\mu}_1 - \widehat{\mu}_2| > 2\delta$ then clearly $\widehat{\mu}' \in \mathrm{Sep}(\delta)$, since the gradient is at most a constant since the function is Lipschitz. Thus assume WLOG that $|\widehat{\mu}_1 - \widehat{\mu}_2| \leq 2\delta \leq \gamma/50$. There are now six cases:

**Case 1:** $\widehat{\mu}_1 \leq \mu_1^* \leq \mu_2^* \leq \widehat{\mu}_2$   This case cannot happen since we assume $|\widehat{\mu}_1 - \widehat{\mu}_2| \leq 2\delta \leq \gamma/50$.

**Case 2:** $\mu_1^* \leq \widehat{\mu}_1 \leq \widehat{\mu}_2 \leq \mu_2^*$   In this case, by Lemma D.6, we know $F$ is negative at $-\infty$ and at $+\infty$. Since clearly $F \geq 0$ when $x \in [\mu_1^*, \mu_2^*]$, by Theorem B.2 and continuity, the discriminator intervals must be of the form $(-\infty, r], [\ell, \infty)$ for some $r \leq \widehat{\mu}_1 \leq \widehat{\mu}_2 \leq \ell$. Thus, the update to $\widehat{\mu}_i$ is (up to a constant factor of $\sqrt{2\pi}$) given by $e^{-(\ell-\widehat{\mu}_i)^2/2} - e^{-(r-\widehat{\mu}_i)^2/2}$. The function $Q(x) = e^{-(\ell-x)^2/2} - e^{-(r-x)^2/2}$ is monotone on $x \in [r, \ell]$, and thus $\widehat{\mu}_i$ must actually move away from each other in this scenario.

**Case 3:** $\mu_1^* \leq \widehat{\mu}_1 \leq \mu_2^* \leq \widehat{\mu}_2$   In this case we must have $|\mu_2^* - \widehat{\mu}_1| \leq 2\delta$ and similarly $|\mu_2^* - \widehat{\mu}_2| \leq 2\delta$. We claim that in this case, the discriminator must be an infinitely long interval $(-\infty, m]$ for some $m \leq \widehat{\mu}_1$. This is equivalent to showing that the function $F(\widehat{\mu}, x)$ has only one zero, and this zero occurs at some $m \leq \widehat{\mu}_1$. This implies the lemma in this case since then the update to $\widehat{\mu}_1$ and $\widehat{\mu}_2$ are then in the same direction, and moreover, the magnitude of the update to $\widehat{\mu}_1$ is larger, by inspection.

We first claim that there are no zeros in the interval $[\widehat{\mu}_1, \widehat{\mu}_2]$. Indeed, in this interval, we have that

$$\sqrt{2\pi} D_{\widehat{\mu}}(x) \geq 2e^{-(\gamma/50)^2/2}$$

$$= 2e^{-\gamma^2/5000}$$

$$\geq 2\left(1 - \frac{\gamma^2}{5000} + O(\gamma^4)\right)$$

$$\geq 2\left(1 - \frac{\gamma^2}{10}\right) ,$$

but

$$\sqrt{2\pi} D_{\mu^*}(x) \leq 1 + e^{-(\gamma - 2\delta)^2/2}$$

$$= 1 + e^{-(49\gamma/50)^2/2}$$

$$\leq 2 - \frac{\gamma^2}{2} .$$

Hence $G_{\widehat{\mu}}(x) \geq G_{\mu^*}(x)$ for all $x \in [\widehat{\mu}_1, \widehat{\mu}_2]$, and so there are no zeros in this interval. Clearly there are no zeros of $F$ when $x \geq \widehat{\mu}_2$, because in that regime $e^{-(x-\widehat{\mu}_i)^2/2} \geq e^{-(x-\mu_i^*)^2/2}$ for $i = 1, 2$. Similarly there are no zeros of $F$ when $x \leq \mu_1^*$. Thus all zeros of $F$ must occur in the interval $[-\mu_1^*, \widehat{\mu}_1]$.

We now claim that there are no zeroes of $F$ on the interval $[\alpha + 10\delta, \widehat{\mu}_1]$, where $\alpha = (\mu_1^* + \widehat{\mu}_1)/2$. Indeed, on this interval, we have

$$\sqrt{2\pi} F(\widehat{\mu}, x) = e^{-(x-\mu_1^*)^2/2} - e^{-(x-\widehat{\mu}_2)^2/2} + e^{-(x-\mu_1^*)^2/2} - e^{-(x-\widehat{\mu}_1)^2/2}$$

$$\leq e^{-(x-\mu_1^*)^2/2} - e^{-(x-\widehat{\mu}_2)^2/2} < 0 ,$$

where the first line follows since moving $\mu_2^*$ to $\widehat{\mu}_1$ only increases the value of the function on this interval, and the final line is negative as long as $x > (\mu_1^* + \widehat{\mu}_2)/2$, which is clearly satisfied by our choice of parameters. By a similar logic (moving $\widehat{\mu}_2$ to $\mu_2^*$), we get that on the interval $[\mu_1^*, \alpha - 10\delta]$, the function is strictly positive. Thus all zeros of $F$ must occur in the interval $[\alpha - 10\delta, \alpha + 10\delta]$.

We now claim that in this interval, the function $F$ is strictly decreasing, and thus has exactly one zero (it has at least one zero because the function changes sign). The derivative of $F$ with respect to $x$ is given by

$$\sqrt{2\pi} \frac{\partial F}{\partial x}(\widehat{\mu}, x)$$

$$= (\mu_1^* - x)e^{-(x-\mu_1^*)^2/2} - (\widehat{\mu}_2 - x)e^{-(x-\widehat{\mu}_2)^2/2}$$

$$+ (\mu_2^* - x)e^{-(x-\mu_2^*)^2/2} - (\widehat{\mu}_2 - x)e^{-(x-\widehat{\mu}_1)^2/2} .$$

By Taylor's theorem, we have

$$(\mu_1^* - x)e^{-(x-\mu_1^*)^2/2} - (\widehat{\mu}_2 - x)e^{-(x-\widehat{\mu}_2)^2/2}$$

$$= -2re^{-\alpha^2/2} + O\left(H_2(\delta)e^{-(r-10\delta)^2/2}\delta^2\right) ,$$

where $H_2$ is the second (probabilist's) Hermite polynomial, and $r = |\mu_1^* - \alpha|$. On the other hand, by another application of Taylor's theorem, we also have

$$(\mu_2^* - x)e^{-(x-\mu_2^*)^2/2} - (\widehat{\mu}_2 - x)e^{-(x-\widehat{\mu}_1)^2/2}$$

$$= O\left(\delta H_2(\delta)e^{-(r-10\delta)^2/2}\right) .$$

Thus, altogether we have

$$\sqrt{2\pi}\frac{\partial F}{\partial x}(\widehat{\mu}, x)$$
$$\leq -2re^{-\alpha^2/2}$$
$$+ O\left(\delta H_2(\delta)e^{-(r-10\delta)^2/2}\right)$$
$$< 0$$

by our choice of $\delta$, and since $r = \gamma/2 > \delta/25$.

**Case 4:** $\widehat{\mu}_1 \leq \mu_1^* \leq \widehat{\mu}_2 \leq \mu_2^*$ This case is symmetric to Case 3, and so we omit it.

**Case 5:** $\mu_1^* \leq \mu_2^* \leq \widehat{\mu}_1 \leq \widehat{\mu}_2$ In this case, we proceed as in the proof of Theorem B.2. If the Gaussians were sufficiently skinny, then by the same logic as in the proof of Theorem B.2, there is exactly one zero crossing. The lemma then follows in this case by Theorem B.1.

**Case 6:** $\widehat{\mu}_1 \leq \widehat{\mu}_2 \leq \mu_1^* \leq \mu_2^*$ This case is symmetric to Case 5.

This completes the proof. □

### D.7 Proof of Lemma 4.6

We also show that no terribly divergent behavior can occur. Formally, we show that if the true means are within some bounded box, then the generators will never leave a slightly larger box.

*Proof of Lemma 4.6.* If $\widehat{\mu} \in B(C)$, then since $f$ is Lipschitz and $\eta$ is sufficiently small, clearly $\widehat{\mu}' \in B(C)$. Thus, assume that there is an $i = 1, 2$ so that $|\widehat{\mu}_i| > C$, and let $\widehat{\mu}_1$ be the largest such $i$ in magnitude. WLOG take $\widehat{\mu}_2 > 0$. In particular, this implies that $\widehat{\mu}_2 > \mu_i^*$ for all $i = 1, 2$. There are now 3 cases, depending on the position of $\widehat{\mu}_1$.

**Case 1:** $\widehat{\mu}_1 \geq \mu_2^*$: Here, as in Case 2 in Lemma 4.5, the optimal discriminator is of the form $(-\infty, r]$ for some $r \leq \widehat{\mu}_1, \widehat{\mu}_2$. In particular, the update step will be

$$\widehat{\mu}_i' = \widehat{\mu}_i - \eta e^{-(r-\widehat{\mu}_i)^2/2} < \widehat{\mu}_i .$$

Thus, in this case our update moves us in the negative direction. By our choice of $\eta$, this implies that $0 \leq \widehat{\mu}_2' < \widehat{\mu}_2'$. Moreover, since $\widehat{\mu}_1 \geq \mu_2^*$, this implies that $|\widehat{\mu}_1| \leq C$, and thus $|\widehat{\mu}_1'| \leq 2C$. Therefore in this case we stay within the box.

**Case 2:** $\mu_1^* \leq \widehat{\mu}_1 \leq \mu_2^*$: As in Case 1, we know that $\widehat{\mu}_1$ cannot leave the box after a single update, as $|\widehat{\mu}_1| \leq C$. Thus it suffices to show that $\widehat{\mu}_2$ moves in the negative direction. By Lemma D.6, we know there is a discriminator interval at $-\infty$, and there is no discriminator interval at $\infty$. Moreover, in this case, we know that $F(\widehat{\mu}, x) \geq 0$ for all $x \geq \widehat{\mu}_2$. Thus, all discriminators must be to the left of $\widehat{\mu}_2$. Therefore, the update to $\widehat{\mu}_2$ is given by

$$\widehat{\mu}_2' = \widehat{\mu}_2$$
$$- \eta \left(e^{-(r_2-\widehat{\mu}_2)^2/2} - e^{-(\ell_2-\widehat{\mu}_2)^2/2} + e^{-(r_1-\widehat{\mu}_2)^2/2}\right) ,$$

for some $r_1 \leq \ell_2 \leq r_2 \leq \widehat{\mu}_2'$. Clearly this update has the property that $0 \leq \widehat{\mu}_2' < \widehat{\mu}_2$, and so the new iterate stays within the box.

**Case 3:** $\widehat{\mu}_1 \leq \mu_2^*$ In this case we must prove that neither $\widehat{\mu}_1$ nor $\widehat{\mu}_2$ leave the box. The two arguments are symmetric, so we will focus on $\widehat{\mu}_2$. Since $\eta$ is small, we may assume that $\widehat{\mu}_2 > 3C/2$, as otherwise $\widehat{\mu}_2'$ cannot leave the box. As in Case 1, it suffices to show that the endpoints of the discriminator intervals are all less than $\widehat{\mu}_2$. But in this case, we have that for all $x \geq \widehat{\mu}_2'$, the value of the true distribution at $x$ is at most $2e^{-(x-C)^2/2}$, and the value of the discriminator is at $e^{-(x-\widehat{\mu}_2)^2/2} \geq e^{-(x-1.5C)^2/2}$. By direct calculation, this is satisfied for any choice of $C$ satisfying $2e^{5C^2/8} < e^{3C^2/4}$, which is satisfied for $C \geq 3$.

□

# E   SINGLE GAUSSIAN

Although our proof of the two Gaussian case implies the single Gaussian case, it is possible to prove the single Gaussian case in a somewhat simpler fashion, while still illustrating several of the high-level components of the overall proof structure. Therefore, we sketch how to do so, in hopes that it provides additional intuition for the proof for a mixture of two Gaussians.

In order to prove convergence, we can use the following.

1. The fact that the gradient is only discontinuous on a measure $0$ set of points.
2. An absolute lower bound on the magnitude of the gradient from below over all points that are not close to the optimal solution that we might encounter over the course of the algorithm
3. An upper bound on how much the gradient can change if we move a certain distance.

Then, as long as we take steps that are small enough to guarantee that the gradient never changes by more than half the absolute lower bound, we will get by Lemma 4.2 that we always make progress towards the optimum solution in function value unless we are already close to the optimal solution.

The proof of these facts is substantially simplified in the single Gaussian case. Suppose we have a true univariate Gaussian distribution with unit variance and mean $\mu^*$, along with a generator distribution with unit variance and mean $\widehat{\mu}$. Then the optimal discriminator for this pair of distributions starts at the midpoint between their means and goes in the direction of the true distribution off to $\infty$ or $-\infty$. Therefore, unless the generator mean is within one step length of the true mean, it cannot move away from the true mean. One can also argue that the gradient of $\widehat{\mu}$ with respect to the optimal discriminator (ie., the gradient of total variation distance) is only discontinuous when $\widehat{\mu} = \mu^*$, and has magnitude roughly $e^{(\widehat{\mu} - (\widehat{\mu} + \mu^*)/2)^2/2}$ for $\widehat{\mu} \neq \mu^*$. This implies the first two items. For the last item, note that the midpoint $e^{z^2/2}$, which implies the gradient is Lipschitz as long as we are not at the optimal solution, which gives bounds on how much the gradient can change if we move a certain distance.

The preceding discussion implies convergence for an appropriately chosen step size, and all this can be made fully quantitative if one works out the quantitative versions of the statements in the preceding argument.

This analysis is simpler the the two Gaussians analysis in several respects. In particular, the proofs of the second two items are substantially more involved and require many separate steps. For example, in the two Gaussian case, the gradient can be $0$ if mode collapse happens, so we have to directly prove both that mode collapse does not happen and that the gradient is large if mode collapse doesn't happen and we aren't too close to the optimal solution, which is a substantially more involved condition to prove. Additionally, the gradient in the two Guassian case does not seem to be Lipschitz away from the optimum like it is in the single Gaussian case. Instead, we will have to use a weaker condition which is considerably more difficult to reason about. This is further complicated by the fact that the optimal discriminators can move in a discontinuous fashion when we vary the generator means.

