# OpenReview forum: "On the limitations of first order approximation in GAN dynamics"
_ICLR.cc/2018/Conference — Invite to Workshop Track_

### Official Review · AnonReviewer3 · 2017-11-19
**An interesting theoretical approach for GAN, but not enough**

**Rating:** 4
**Confidence:** 4

**Review:**

Although GAN recently has attracted so many attentions, the theory of GAN is very poor. This paper tried to make a new insight of GAN from theories and I think their approach is a good first step to build theories for GAN.

However, I believe this paper is not enough to be accepted. The main reason is that the main theorem (Theorem 4.1) is too restrictive.

1.	There is no theoretical result for failed conditions.
2.	To obtain the theorem, they assume the optimal discriminator. However, most of failed scenarios come from the discriminator dynamics as in Figure 2.
3.	The authors could make more interesting results using the current ingredients. For instance, I would like to check the conditions on eta and T to guarantee d_TV(G_mu*, G_hat{mu})<= delta_1 when |mu*_1 – mu*_2| >= delta_2 and |hat{mu}_1 – hat{mu}_2| >= delta_3. In Theorem 4.1, the authors use the same delta for delta_1, delta_2, delta_3. So, it is not clear which initial condition or target performance makes the eta and T.

---

> ### Author Response · Authors · 2018-01-04
> **Author response**
>
> We thank the reviewer for appreciating our approach for building a theory for GANs. We now address the concerns:
>
> 1) We do not provide theoretical results for the failed conditions because the experiments already demonstrate convincingly that the first-order methods for training discriminators have serious deficiencies in our model. Moreover, in the supplementary material, we discuss specific ways in which the first-order approach fails. For instance, Figure 2 shows that for most initial generator states, less than 20% of the discriminator configurations give rise to first-order dynamics that successfully learn the unknown distribution.
>
> 2) Our convergence results are indeed only for the optimal discriminator. However, this is a necessity, because (as the reviewer points out) the first order dynamics often do not converge. Therefore, it is impossible to even hope for a general convergence result in this setting. In fact, we view demonstrating that the first-order dynamics can fail in such a systematic way an important contribution of our paper. In particular, this hints towards a fundamental separation between optimal and first order dynamics for training GANs, and the need to understand what we can and cannot achieve when we rely on first-order methods for training
>
> 3) We agree with the reviewer that it would be interesting to understand the relationships between the parameters at a more fine-grained level. However, since it did not seem to change the qualitative message of the results, we chose not to optimize parameters in favor of simplicity of exposition. In the updated version, we will flesh out said relationships more explicitly.

---

### Official Review · AnonReviewer1 · 2017-11-24
**too simple a model?**

**Rating:** 5
**Confidence:** 3

**Review:**

The authors proposes to study the impact of GANS in two different settings:
1. at each iteration, train the discriminator to convergence and do a (or a few) gradient steps for updating the generator
2. just do a few gradient steps for the discriminator and the generator
This is done in a very toy example: a one dimensional equally weighted mixture of two Gaussian distributions.

Clarity: the text is reasonably well written, but with some redundancy (e.g. see section 2.1) , and quite a few grammatical and mathematical typos here and there. (e.g. Lemma 4.2., $f$ should be $g$, p7 Rect(0) is actually the empty set, etc..)

Gaining insights into the mechanics of training GANs is indeed important. The authors main finding is that, in this very particular setting, it seems that training the discriminator to convergence leads to convergence. Indeed, in real settings, people have tried such strategies for WGAN for examples. For standard GANs, if one adds a little bit of noise to the labels for example, people have also reported good result for such a strategy (although, without label smoothing, this will indeed leads to problems).

Although I have not checked all the mathematical fine details, the approach/proof looks sound (although it is not at all clear too me why the choice of gradient step-sizes does not play a more important roles the the stated results). My biggest complain is that the situation analyzed is so simple (although the convergence proof is far from trivial) that I am not at all convinced that this sheds much light on more realistic examples. Since this is the main meat of the paper (i.e. no methodological innovations), I feel that this is too little an innovation for deserving publication in ICLR2018.

---

> ### Author Response · Authors · 2018-01-04
> **Author response**
>
> The main concern of the reviewer is that our model is simplistic, and that our insights might not transfer to more realistic settings. While we agree that our model is simple, we argue that it is a necessary first step. In fact, we believe that this simplicity is an advantage of our paper.
>
> Currently, our understanding of GAN training is in its infancy. While a large number of GAN variants has been proposed, basic questions about the convergence of even simple GANs are still unanswered. Hence it is crucial to begin a principled and rigorous investigation of GAN dynamics to demystify GAN training. From this point of view, studying common methods in simple settings is an important first step: if we do not understand basic principles (such as the impact using first-order approximations when training  discriminators) even in such simple settings, there is no hope for gaining such understanding in more complex setups. Following this viewpoint, the absence of methodological novelty in our paper is intentional so we can highlight fundamental aspects of standard GAN training in a rigorous fashion.
>
> Indeed, we have shown that the convergence analysis for optimal discriminator dynamics is already highly non-trivial, even in a simple model. Moreover, we have empirically demonstrated that the natural first order GAN dynamics fail to converge for this model. Any future theory for more sophisticated GANs will have to handle these phenomena as a special case (or exclude our setup via stringent assumptions). Hence we believe that rigorously investigating our simple model is an important contribution.
>
> We thank the reviewer for finding the typos. We will correct them in the final version of the paper.

---

### Official Review · AnonReviewer2 · 2017-12-02
**Good (but limited) step towards understanding dynamics of adversarial training in GANs**

**Rating:** 7
**Confidence:** 3

**Review:**

Summary:

This paper studies the dynamics of adversarial training of GANs for Gaussian mixture model. The generator is a mixture of two Gaussians in one dimension. Discriminator is union of two intervals. Synthetic data is generated from a mixture of two Gaussians in one dimension. On this data, adversarial training is considered under three different settings depending on the discriminator updates: 1) optimal discriminator updates, 2) standard single step gradient updates, 3) Unrolled gradient updates with 5 unrolling steps.

The paper notices through simulations that in a grid search over the initial parameters of generator optimal discriminator training always succeeds in recovering the true generator parameters, whereas the other two methods fail and exhibit mode collapse. The paper also provides theoretical results showing global convergence for the optimal discriminator updates method.



Comments:
1) This is an interesting paper studying the dynamics of GANs on a simpler model (but rich enough to display mode collapse). The results establish the standard issues noticed in training  GANs. However no intuition is given as to why the mode collapse happens or why the single discriminator updates fail (see for ex. https://arxiv.org/abs/1705.10461)?

2) The proposed method of doing optimal discriminator updates cannot be extended when the discriminator is a neural network. Does doing more unrolling steps simulate this behavior? What happens in your experiments as you increase the number of unrolling steps?

3) Can you write the exact dynamics used for Theorem 4.1 ? Is the noise added in each step?

4) What is the size of the initial discriminator intervals used for experiments in figure 2?

---

> ### Author Response · Authors · 2018-01-04
> **Author response**
>
> We thank the reviewer for the positive feedback.
>
> 1) Regarding intuition: In the supplementary material, we highlight a specific failure case that we have observed in our model (the so-called “ discriminator collapse”). At a high level, the discriminator is often incentivized to decrease its representational power in order to increase its current accuracy when using first-order updates. This causes the discriminator to fail to adapt later on (when the generator changes) and can lead to training failure.
>
> 2) The reviewer asks if unrolling steps can simulate the optimal dynamics. As reported in the paper, we also experimented with unrolling steps. We found that these unrolling steps did not avoid the pathological behaviors of using single gradient step updates. Our results suggest, in fact, that no dynamics solely based on first order updates can avoid these pathologies.
>
> 3) The dynamics in Theorem 4.1 are exactly as stated in equation (7) except we add (very small) Gaussian noise at each step. While we believe that this is unnecessary (at least when we randomly initialize the parameters), our current proof requires this modification. In our experiments, the dynamics as written in Theorem 4.1 (i.e., without noise) always converged.
>
> 4) In Figure 2, the initial intervals have endpoints which are drawn iid from the interval [-4, 4] and then sorted. We remark that we did not find any qualitative change in our experiments  when we used different choices of initial intervals.

---

### Decision · Program_Chairs · 2018-01-29
**ICLR 2018 Conference Acceptance Decision**

**Decision:**

Invite to Workshop Track

**Comment:**

All the reviewers agree that the paper is studying an important problem and makes a good first step towards understanding learning in GANs. But the reviewers are concerned that the setup is too simplistic and not relevant in practical settings. I recommend the authors to carefully go through reviews and to present it at the workshop track. This will hopefully foster further discussions and lead to results in more practically relevant settings.